# Autophagosomes anchor an AKAP11-dependent regulatory checkpoint that shapes neuronal PKA signaling

Ashley Segura-Roman[1,3], Y Rose Citron[1,3], Myungsun Shin[2], Nicole Sindoni[2], Alex Maya-Romero [1], Simon Rapp [1], Claire Goul[1], Joseph D Mancias[2] & Roberto Zoncu [1✉]

## Abstract

Protein Kinase A (PKA) is regulated spatially and temporally via scaffolding of its catalytic (Cα) and regulatory (RI/RII) subunits by the A-kinase-anchoring proteins (AKAP). By binding to an AKAP11 scaffold, PKA engages in poorly understood interactions with autophagy, a key degradation pathway for neuronal cell homeostasis. Mutations in AKAP11 promote schizophrenia and bipolar disorders (SZ-BP) through unknown mechanisms. Here, through proteomic-based analyses of immunopurified lysosomes, we identify the Cα-RIα-AKAP11 holocomplex as a prominent autophagy-associated protein-kinase complex. AKAP11 scaffolds Cα-RIα interaction with the autophagic machinery via its LC3-interacting region (LIR), enabling both PKA regulation by upstream signals, and its autophagy-dependent degradation. We identify Ser83 on the RIα linker-hinge region as an AKAP11-dependent phospho-residue that modulates RIα-Cα binding to the autophagosome and cAMP-induced PKA activation. Decoupling AKAP11-PKA from autophagy alters downstream phosphorylation events, supporting an autophagy-dependent checkpoint for PKA signaling. Ablating AKAP11 in induced pluripotent stem cell-derived neurons reveals dysregulation of multiple pathways for neuronal homeostasis. Thus, the autophagosome is a platform that modulates PKA signaling, providing a possible mechanistic link to SZ/BP pathophysiology.

**Keywords** Protein kinase A; AKAP11; Autophagy; Phosphoproteomics; Signaling
**Subject Categories** Autophagy & Cell Death; Signal Transduction

## Introduction

PKA is a prominent Ser/Thr kinase that transduces cyclic adenosine monophosphate (cAMP)-mediated signals from many different hormones and neurotransmitters to instruct biological responses ranging from regulation of glucose and lipid metabolism in peripheral tissues, to memory formation and consolidation in the brain (Kandel, 2012; London et al, 2020; Monterisi and Zaccolo, 2017; Tomek and Zaccolo, 2023). In its inactive state, PKA exists as a holoenzyme composed of a homodimer of two regulatory (R) subunits, each of which in turn holds and inhibits two catalytic subunits. Binding of cAMP to two nucleotide binding domains (NBD) on each R subunit induces a conformational change that frees up the C-subunit active site, enabling phosphorylation of a wide variety of downstream substrates (Kim et al, 2007; Taylor et al, 2013; Wu et al, 2007).

Four subtypes of R-subunits, RIα, RIβ, RIIα, and RIIβ, form as many distinct homodimers In addition to C-subunits each R-dimer, in turn, can bind to several A-kinase anchoring proteins (AKAPs). AKAP proteins organizes PKA signaling into a distributed intracellular network, where microdomains of PKA holoenzymes, scaffolded to different subcellular compartments, perform specific regulatory actions upon local elevation of cAMP levels (Scott et al, 2013). Detailed mechanistic studies of AKAP proteins have conclusively demonstrated their ability to link PKA activity to specific organelles and cellular compartments, including the plasma membrane, nuclear envelope, mitochondria, Golgi apparatus, lipid droplets and centrosome (Monterisi and Zaccolo, 2017; Omar and Scott, 2020).

AKAP11 (also known as AKAP220) is among the least well understood AKAPs. This 220-KDa PKA-scaffolding protein has been implicated in processes such as actin turnover (Logue et al, 2011), microtubule dynamics (Logue et al, 2011), and osmoregulation (Whiting et al, 2016). AKAP11 has two binding sites for the regulatory subunits, one which binds to both RIα and RIIα, whereas the second one primarily binds to RIIα (Whiting et al, 2015). Along with the PKA holoenzyme, AKAP11 scaffolds glycogen synthase kinase (GSK) IIIβ, possibly allowing cross-regulation between the two kinases (Tanji et al, 2002; Whiting et al, 2015). AKAP11 also binds to protein phosphatase 1 (PP1), possibly as part of an integrated system to rapidly activate and then shut down downstream substrates for both PKA and GSKIII-β (Schillace and Scott, 1999).

[1]Department of Molecular and Cell Biology, University of California at Berkeley, Berkeley, CA 94720, USA. [2]Division of Radiation and Genome Stability, Department of Radiation Oncology, Dana-Farber Cancer Institute, Harvard Medical School, Boston, MA 02215, USA. [3]These authors contributed equally: Ashley Segura-Roman, Y Rose Citron.
✉E-mail: rzoncu@berkeley.edu

Recently, polymorphisms in the AKAP11 gene sequence that lead to protein truncation and loss-of-function have been associated with increased risk for schizophrenia (SCZ) and bipolar disorder (BP) (Herzog et al, 2023; Liu et al, 2023; Palmer et al, 2022). A deep-proteomic study comparing brain tissue samples from SCZ and BP patients with synaptosomal preparations from AKAP11-knockout mice revealed shared alterations in the abundance of protein classes that are important for brain function, including mitochondria, vesicle trafficking and tethering (Aryal et al, 2023). However, the molecular links between AKAP11 deficiency and neuronal dysfunction that leads to SCZ and BP remain unknown.

Several reports have implicated AKAP11 in directing PKA to specific vesicle populations. AKAP11 was shown to recruit RIα to multivesicular bodies (MVBs), but only when RIα is not bound to Cα such as in high cAMP concentrations (Day et al, 2011). More recently, AKAP11 was proposed to function as a selective autophagy receptor that, by interacting with ATG8-family proteins, causes selective RIα capture and degradation into autophagosomes, while sparing Cα from degradation (Deng et al, 2021; Zhou et al, 2024).

Autophagy is a vesicular catabolic pathway that singles out proteins, macromolecules and organelles for capture within double-membraned autophagosomes, which subsequently deliver their cargo to lysosomes for degradation (Melia et al, 2020; Yamamoto et al, 2023). Recognition of autophagic substrates relies on a receptor-adaptor system (Lamark and Johansen, 2021; Vargas et al, 2023). Autophagic adaptors of the ubiquitin-like ATG8-family: MAP1LC3A/B/C, GABARAP and GABRAPBL1/2, are inserted in the autophagosomal membrane via their C-terminal conjugation to phosphatidylethanolamine (PE). ATG8-family adaptors recognize conserved linear motifs known as LC3-interacting regions (LIRs) on selective autophagic receptors, which in turn recruit various cargo to the nascent autophagosome (Ichimura et al, 2008; Kirkin and Rogov, 2019).

Consistent with the critical roles of autophagy in nutrient homeostasis and cellular quality control, this pathway is subjected to regulation by master nutrient-sensing kinases such as mTORC1, which negatively regulates the autophagosome-nucleating ULK1 complex under high nutrients, and AMPK, which stimulates several autophagic regulatory complexes during energy shortage (Kaur and Debnath, 2015; Shin and Zoncu, 2020). Conversely, emerging evidence suggests that autophagy can exert feedback regulation on upstream signaling pathways: for example, nutrients generated by autophagic breakdown of macromolecules can reactivate mTORC1 and shut down AMPK signaling during recovery from starvation (Liu and Sabatini, 2020; Shin and Zoncu, 2020). However, unlike lysosomes and mitochondria, which carry out well-recognized signaling functions on their cytoplasmic face that complement their internal (luminal) activities (Chandel, 2015; Shin and Zoncu, 2020), whether autophagosomes can act as organizers of cellular signaling at any point during their life cycle remains unclear.

The relationship between PKA signaling and autophagy is complex and not well understood. Phosphoproteomic studies in yeast and human cells identified many autophagic regulators and effectors as PKA substrates, including the ATG1/ULK1 initiation complex, the ATG8 and ATG12 conjugation systems, as well as accessory factors and adaptors (Budovskaya et al, 2005; Grisan et al, 2021; Isobe et al, 2017; Stephan et al, 2009). PKA-dependent phosphorylation of LC3B and ATG16L1 were shown to inhibit autophagosome formation in neurons and endothelial cells, respectively, suggesting that, like mTORC1, PKA may inhibit autophagy (Cherra et al, 2010; Zhao et al, 2019). Conversely, functional genomic screens in yeast identified genes involved in each step of autophagy, from induction to substrate breakdown, as candidate negative regulators of PKA signaling (Filteau et al, 2015).

The findings that AKAP11 promotes autophagic degradation of RIα has led to a model in which selective RIα elimination increases the ratio of free C-subunit, thereby promoting PKA signaling (Deng et al, 2021; Zhou et al, 2024). This mechanism was proposed to potentiate C-mediated phosphorylation of PKA-cAMP response element-binding (CREB) transcription factor, boosting mitochondrial metabolism and conferring resistance to glucose deprivation (Deng et al, 2021). Along similar lines, a further study proposed that autophagy-dependent elimination of RIβ in neurons frees up Cα to enhance PKA signaling at the synapse (Overhoff et al, 2022).

However, these models need to be evaluated in light of established features of PKA regulation. First, the R-subunit does not play a purely inhibitory role toward the C-subunit. On the contrary, by scaffolding C to different AKAP proteins, R helps place the former in proximity to numerous organelle-specific substrates (Means et al, 2011; Monterisi and Zaccolo, 2017; Omar and Scott, 2020; Pidoux et al, 2011). Thus, decreasing the stoichiometry of R- to C-subunits would be expected to have mixed effects on downstream signaling, favoring some phosphorylation events while decreasing others. Second, recent evidence suggests that, at physiological cAMP levels found within the cell, C-subunits do not completely separate from the R-subunit-AKAP complex. Instead, the three components remain bound in an open, signaling-competent conformation that is capable of substrate recognition and phosphorylation (Smith et al, 2013, 2017). Thus, autophagic capture of AKAP11-RIα seems unlikely to completely spare C-subunits from degradation.

To achieve a fine-grained understanding of how autophagy regulates PKA, and possibly other multiprotein signaling complexes, we combined immunoisolation and proteomic-based profiling of lysosomes from autophagy competent and deficient cells with bioinformatic analysis of protein–protein interaction datasets. These experiments identify the PKA holoenzyme, primarily composed of the RIα and Cα subunits and AKAP11, as a kinase complex prominently associated with autophagy. In agreement with previous reports, AKAP11 bridges RIα to LC3B and GABARAP via its LIR motif. However, we find that Cα is also scaffolded to LC3B and GABARAP via its association with RIα and AKAP11. Crucially, association with the autophagic machinery modulates PKA-dependent signaling, both by co-scaffolding the PKA holoenzyme with upstream regulators, and by promoting its degradation. In particular, phosphoproteomic analysis in iPSC-derived neurons reveals AKAP11-dependent RIα phosphorylation on a newly identified regulatory site, Ser83; accordingly, depleting AKAP11 or disabling its interaction with the autophagy machinery led to dysregulation of downstream signaling programs linked to neuronal cell homeostasis. Collectively, our data point to the autophagosome as a central signaling hub that plays a critical role in shaping PKA-dependent signaling responses via AKAP11. Alteration of these signaling programs upon AKAP11 loss may contribute to the pathophysiology of SZ/BP.

# Results

## Proteomic and bioinformatic identification of multiprotein autophagic substrates

We reasoned that signaling complexes that interact tightly with the autophagic machinery may be subjected, at least in part, to autophagy-dependent degradation. To identify these autophagy-associated signaling complexes, we carried out lysosome immunoisolation and proteomic-based profiling from control HEK-293T cells versus HEK-293T cells deleted for an essential autophagic regulator, the E1-like factor ATG7 (Abu-Remaileh et al, 2017; Davis et al, 2021) (Figs. 1A and EV1A,B). To unambiguously distinguish lysosomal resident proteins from bona fide substrates, we compared lysosomal immunoprecipitates from DMSO-treated control and ATG7-KO cells, versus cells treated with the protease inhibitor cocktail, leupeptin + pepstatin (L + P) (Davis et al, 2021). Proteins that showed selective accumulation in L + P-treated control lysosomes included canonical autophagic receptors CALCOCO2, NBR1, TAX1BP1 and NCOA4, along with transmembrane and extracellular proteins that reach the lysosome via endocytic uptake and endosomal trafficking (Dataset EV1). As expected, autophagic receptors were depleted in lysosomes immunopurified from ATG7-KO cells (Dataset EV1), establishing a clear pattern for recognition of bona fide autophagic substrates.

Gene Ontology (GO) analysis showed differential accumulation of several classes of putative autophagic substrates in lysosomes from control, L + P treated cells versus ATG7-KO, L + P-treated cells. The most autophagy-dependent substrate categories included factors involved in protein folding, transport and quality control (Figs. 1B and EV1C).

We next analyzed the proteomic data through a custom-written bioinformatic pipeline that clusters proteins based on physical interaction (BioGRID database) (Figs. 1A and EV1B). The resulting network view identified several multiprotein complexes as novel substrates of autophagic degradation (Figs. 1C and EV2A). We identified complexes involved in protein quality control, including the proteasome (12 subunits detected with $P$ value = 2.05E-05) and the CCT/TRiC chaperone complex (7 subunits) (Fig. 1C; Dataset EV1). Also detected was the HSP90 chaperone complex, including HSP90α class A and B, their p23/PTGES3 and FKBP4 cofactors, as well as HSP70, which, while not stably bound to HSP90, functions in close association with HSP90 on substrate proteins (Moll et al, 2022) (Fig. 1C; Dataset EV1). The enrichment of protein quality control factors likely reflects their abundant levels in the cell, as well as their degradation along with   client proteins that failed to correctly fold or assemble.

Our pipeline also identified multiprotein complexes involved in signal transduction, specifically the catalytic and one regulatory subunits of protein phosphatase 2 (PPP2CA and PPP2R1A, respectively), and the PKA enzyme complex composed of the Cα catalytic subunit, the RIα and RIIα regulatory subunits, and the AKAP11 (also known as AKAP220) scaffolding protein (Fig. 1C; Dataset EV1).

## The PKA holoenzyme is a substrate for autophagic degradation

To independently validate the proteomic data, we immunoisolated lysosomal samples from control and ATG7-KO HEK-293T cells, as well as from cells deleted for FIP200, a component of the ULK1

complex that is essential for autophagy initiation (Yamamoto et al, 2023). We carried out lyso-IP both in baseline conditions and following treatment with the vacuolar H$^+$ATPase (v-ATPase) inhibitor, Bafilomycin A1 (BafA1), which, like L + P, inhibits substrate breakdown within the lysosomal lumen (Fig. 1D). Immunoblotting of these samples confirmed the autophagy-dependent degradation of the multiprotein complex components detected by mass spectrometry, including the proteasome, HSP90, CCT/TRiC, PP2A (Fig. 1D). Supporting autophagic degradation of the PKA holoenzyme, RIα, RIIα, Cα and AKAP11 were clearly detected in lysosomes from control cells treated with BafA1, but not in lysosomes from BafA1-treated ATG7- and FIP200-KO cells (Fig. 1D). In light of recently proposed models of autophagy-dependent regulation of PKA (Deng et al, 2021; Overhoff et al, 2022; Zhou et al, 2024), and the recently discovered association of AKAP11 mutations with SCZ and BP (Herzog et al, 2023; Liu et al, 2023; Palmer et al, 2022), we decided to further investigate the Cα-R1α−AKAP11 complex as a potential autophagy interactor and substrate.

To gain a quantitative understanding of PKA-AKAP11 capture by autophagy, we visualized the lysosomal mass spectrometry data (Dataset EV1) as volcano plots. These plots showed that RIα is among the most enriched proteins by L + P treatment, on par with canonical autophagic receptors such as TAX1BP1 and p62, whereas Cα and AKAP11 were stabilized to a lesser degree (Fig. 2A). The same relative enrichment was found when comparing lysosomes from L + P-treated wild-type versus ATG7-deleted cells (Fig. 2B).

Autophagy-dependent capture of the PKA holoenzyme was also evident in immunofluorescence-based experiments. In cells stably expressing FLAG-tagged RIα, blocking lysosomal proteolysis with BafA1 led to pronounced accumulation of FLAG-R1α in LAMP2-positive lysosomes of control, but not ATG7-deleted, HEK-293T cells (Fig. 2C,D). Similarly, in HEK-293T cells in which endogenous Cα is C-terminally tagged with mNeon (Cho et al, 2022), treatment with BafA1 led to pronounced accumulation of mNeon-Cα in LAMP2-positive lysosomes that was abolished by shRNA-mediated knock down of FIP200, consistent with autophagy-dependent capture and degradation of Cα (Fig. 2E,F).

In conclusion, converging evidence from organelle proteomics and immunolocalization support autophagic capture and lysosomal degradation of both the regulatory and catalytic PKA subunits via an ATG7- and FIP200-dependent mechanism.

## AKAP11 mediates capture of the PKA holoenzyme

It was previously reported that AKAP11 promotes autophagic degradation of a fraction of the RIα pool (Deng et al, 2021; Zhou et al, 2024). However, our detection of Cα in immunopurified lysosomes, as well as by co-localization analysis in intact cells, indicate that the autophagic machinery interacts with not just RIα, but with the PKA holoenzyme (Figs. 1C,D and 2E,F). To verify that autophagic capture of Cα is also AKAP11-dependent, we stably expressed the lysosomal affinity tag in both control and AKAP11-deleted HEK-293T cells, and immunoblotted lysosomal samples for multiple PKA proteins. This analysis clearly showed that both Cα and RIα were strongly depleted from lysosomes immunopurified from AKAP11-KO cells (Fig. 3A).

Whereas the R−AKAP interaction is relatively stable (in the nanomolar range), Cα progressively dissociates from the R-AKAP

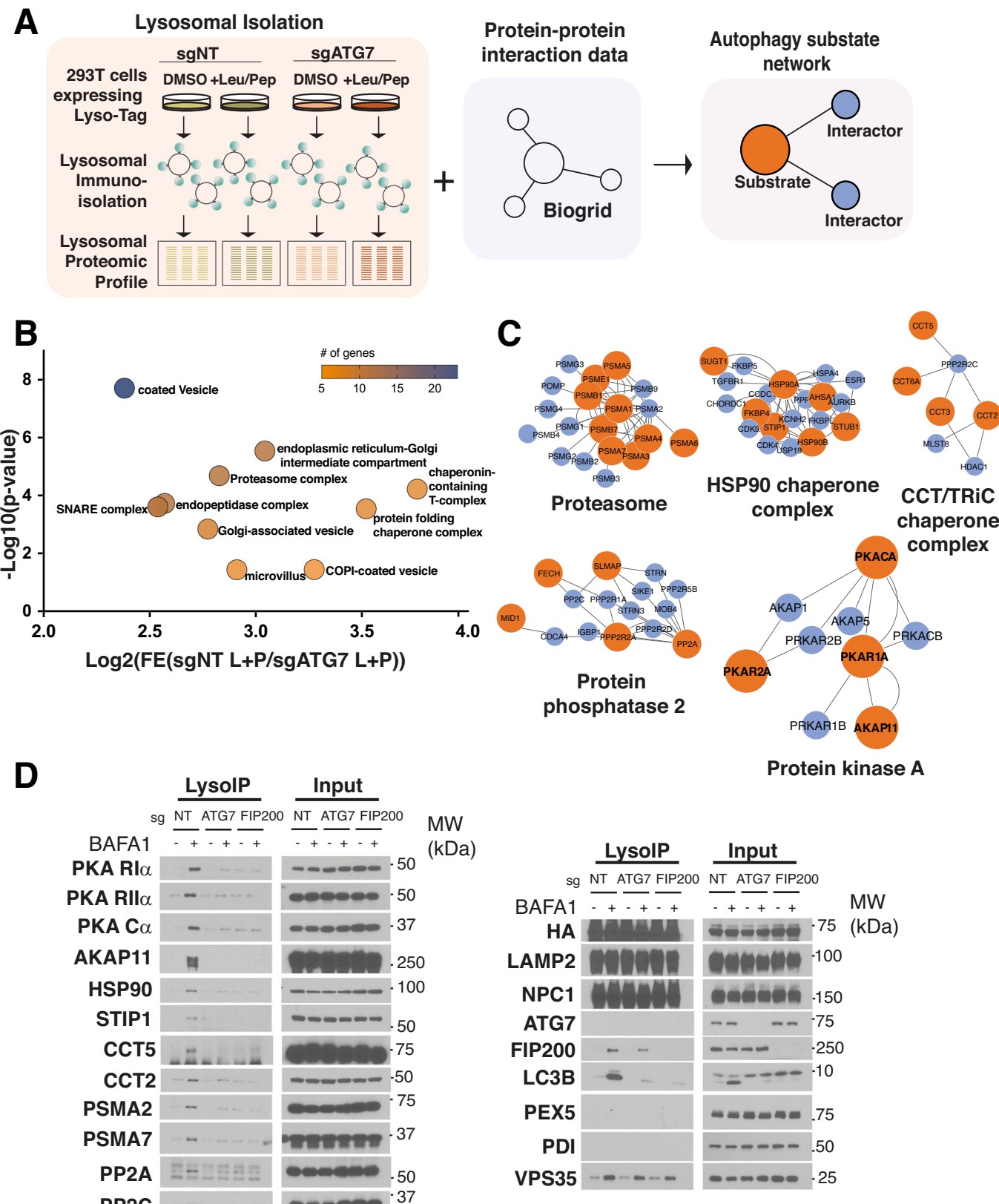

**Figure 1.   Multiprotein complexes are autophagic substrates.**

(A) Summary chart of the workflow for identification of autophagy-dependent degradation of multi-protein complexes. HEK293T sgNT or sgATG7 cells were treated with 20 μM Leupeptin + 20μM Pepstatin or DMSO for 24 h prior to lysosomal immunoprecipitation. Resulting proteomic analysis from proteins classified as autophagic-dependent lysosomal substrates were integrated into a network using protein–protein interaction data from the Biogrid database. Four citations were required for protein–protein interaction. (B) Plot of "cellular component slim" GO-terms enriched in wild-type lysosomes compared to autophagy-null lysosomes in the Leupeptin + Pepstatin condition. (C) Highlighted multi-protein complexes from protein network analysis. Orange nodes represent detected autophagy-dependent substrates. Blue nodes represent cited physical protein interactors. Proteins belonging to the proteasome, HSP90 Chaperone complex, CCT/TRiC chaperone complex, protein phosphatase 2, or protein kinase A complexes are enriched as autophagy-dependent lysosomal substrates. (D) Immunoblots of lysosomal immunoprecipitation and corresponding input from HEK293T sgNT, sgATG7 or sgFIP200 after treatment with 500 nM bafilomycin A1 (BafA1) for 5 h to block lysosomal degradation of autophagic substrates. Source data are available online for this figure.

complexes as cytoplasmic concentrations of cAMP increase (Bock et al, 2024). However, recent structural and biochemical evidence indicates that concentrations of cAMP that are sufficient to activate PKA signaling do not cause complete Cα dissociation from the R-AKAP complex. Instead, the AKAP-holoenzyme remains associated while adopting a range of dynamic conformations that are compatible with substrate engagement and phosphorylation (Smith et al, 2013, 2017). In line with this idea, HA-tagged AKAP11 co-immunoprecipitated both RIα and Cα in full media, under which conditions PKA is catalytically active. To achieve complete separation of Cα from the RIα−AKAP11 complex, we had to treat cells with the adenylyl cyclase activator, forskolin, at >1 μM concentrations, which lead to supra-physiological cAMP levels in the cell (Smith et al, 2017) (Fig. 3B).

To be captured into autophagosomes, substrate proteins must interact with Atg8-family proteins LC3A/B/C or GABARAP, either via an LC3-interacting region (LIR) within their amino acid sequence, or indirectly through a selective autophagic receptor (Adriaenssens et al, 2022; Kirkin and Rogov, 2019). In co-immunoprecipitation (co-IP) experiments, GFP-tagged LC3B and GABARAP bound to RIα, Cα and AKAP11 with relatively high strength (Fig. 3C). The RIα and Cα interactions were largely abolished when using point mutants in the hydrophobic groove of LC3 and GABARAP that impair recognition of canonical LIR motifs (P52A and Y46A, respectively) (Ichimura et al, 2008; Kirkin and Rogov, 2019) (Fig. 3C, top). Notably, the same hydrophobic groove mutations in LC3 and GABARAP only partially abolished binding to endogenous AKAP11, suggesting additional, LIR-independent modes of interaction between AKAP11 and Atg8 proteins (Fig. 3C, top). Supporting the requirement for AKAP11 in autophagic capture of the PKA holoenzyme, GFP-LC3B and GFP-GABARAP failed to immunoprecipitated endogenous RIα and Cα when expressed in AKAP11-deleted cells, whereas their binding to p62/SQSTM1 was unaffected by lack of AKAP11 protein (Fig. 3C, bottom).

Similar to HA-AKAP11, GFP-LC3B co-immunoprecipitated both RIα and Cα in full media conditions. Stimulating GFP-LC3B-expressing cells with increasing concentrations of forskolin progressively reduced the amount of Cα immunoprecipitated by GFP-LC3B, consistent with Cα interacting with LC3B indirectly, via its association with RIα (Fig. 3D). As for the AKAP11 co-IP experiments (Fig. 3B), Cα fully separated from LC3B only at supraphysiological concentrations of forskolin ( >1 μM) (Fig. 3D).

Thus, under steady-state conditions autophagy captures and degrades the AKAP11-bound pool of PKA holoenzyme via association of both RIα and Cα with AKAP11 and LC3/GABARAP.

## AKAP11 binding to LC3 and R1α is required for autophagic degradation of the PKA holoenzyme

We next tested the mechanisms underlying autophagic capture of the AKAP11-containing PKA holoenzyme. Human AKAP11 contains a putative LIR motif, WSNL, at a.a. 1736–1739 (Deng et al, 2021). Wild-type, V5-tagged AKAP11 (stably expressed in AKAP11-deleted HEK-293T cells) was readily detected within LAMP2-positive lysosomes upon BafA1-treatment; in contrast, mutating the WSNL motif to four alanines caused the resulting AKAP11[4ALIR] mutant to be largely excluded from lysosomes and instead maintain a diffuse staining pattern upon BafA1 treatment (Fig. 4A,B).

As the 1736–1739 LIR motif of AKAP11 was shown to be required for AKAP11-mediated RIα degradation (Deng et al, 2021), we sought to determine its requirement for autophagic capture of Cα as well. For this, we ablated AKAP11 in HEK-293T cells in which endogenous Cα is tagged with mNeon, and reconstituted them with stably expressed wild-type AKAP11 or with the AKAP11[4ALIR] mutant (Fig. EV3A). AKAP11[4ALIR] bound to endogenous Cα and RIα with identical strength to the wild-type protein (Fig. 3B).

Immunofluorescence analysis showed that ablating AKAP11 blocked lysosomal localization of mNeon-Cα in BafA1-treated cells (Fig. 4C,D). mNeon-Cα – LAMP2 co-localization was restored by re-expressing AKAP11[WT], but not by the AKAP11[4ALIR] mutant (Fig. 4C,D). Thus, while dispensable for the integrity of the Cα−RIα-AKAP11 complex, the LIR motif of AKAP11 appears critical for autophagic capture of the PKA holoenzyme.

Because AKAP11 and RIα form a stable complex, we also examined potential roles for RIα in promoting autophagic capture of the holoenzyme. We generated RIα-deleted HEK-293T cells via CRISPR-Cas9, and carried out lysosome affinity purification and immunoblotting from these cells. As expected, deleting RIα prevented accumulation of Cα in lysosomes (Fig. 4E). Interestingly, the levels of lysosomal AKAP11 were also suppressed in lysosomes from RIα-deleted cells, suggesting that AKAP11 must be bound to RIα in order for its capture by the autophagic machinery to occur (Fig. 4E). To independently corroborate these data, we deleted the two RI binding sites, AA 615–628 and AA 1650–1663, from AKAP11 (AKAP11 ΔS1S2), while leaving the LIR motif intact. Consistent with the lysoIP data, AKAP11 ΔS1S2 co-localized with the lysosome to a lesser degree than AKAP11 WT, though to a higher than the AKAP11[4ALIR] mutant (Fig. EV3B,C).

Together, these data show that AKAP11 and RIα cooperatively scaffold the entire Cα− RIα-AKAP11 complex to autophagosomes via LIR-dependent interactions between AKAP11 and the autophagic machinery.

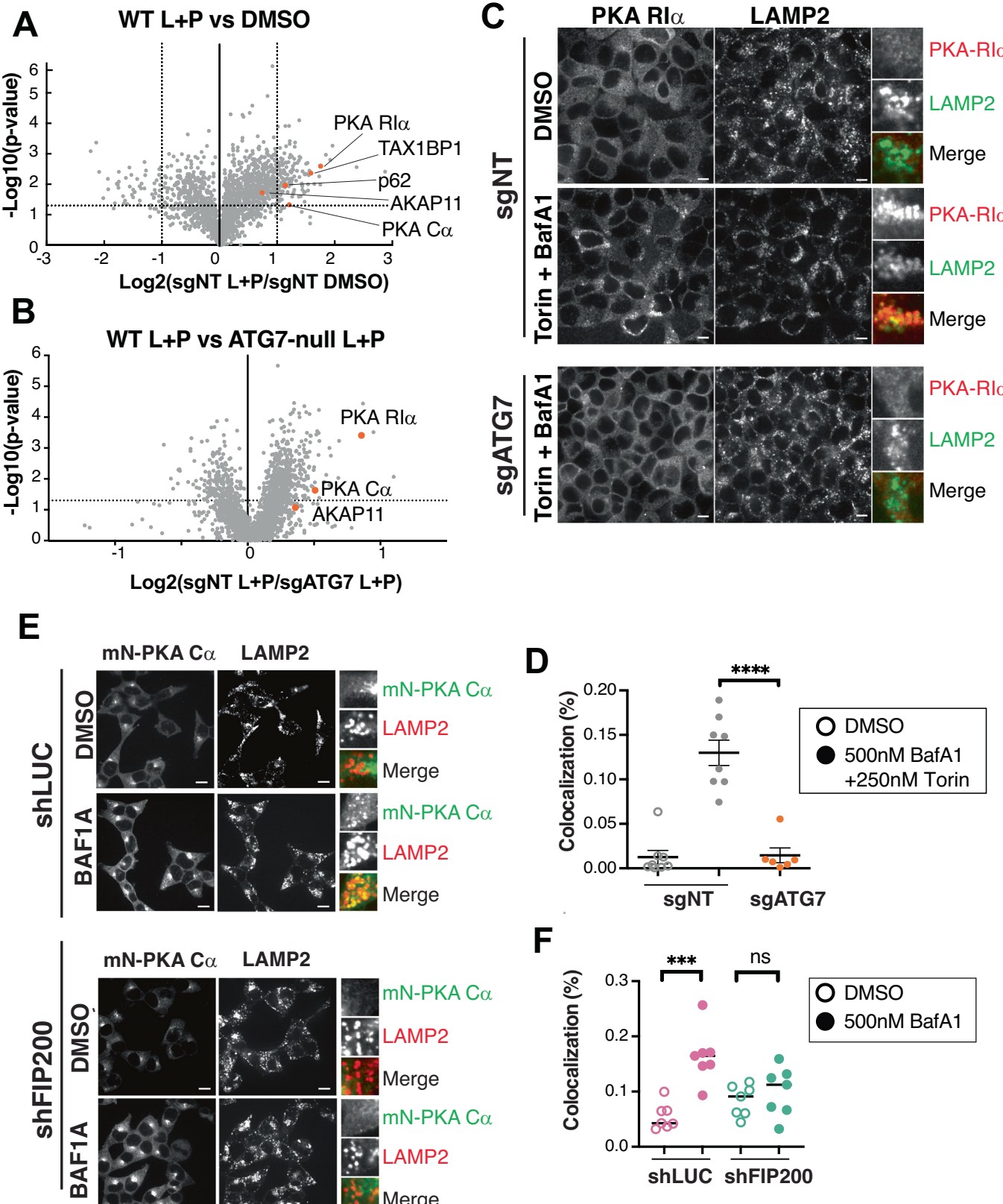

**A** WT L+P vs DMSO

**B** WT L+P vs ATG7-null L+P

**C** PKA RIα LAMP2

**D**

**E** mN-PKA Cα LAMP2

**F**

**Figure 2. The PKA holoenzyme is degraded via autophagy.**

(A, B) Proteomic analysis of Lyso-immunoprecipitation (LysoIP) from HEK293T sgNT and sgATG7. $n = 3$, independent biological replicates for all conditions. *P* values calculated using two-tailed unpaired *t* test. (A) Volcano plots show the ratio of 20 µM Leupeptin + 20µM Pepstatin (L + P) to DMSO treated for sgNT lysoIP samples. (B) Volcano plots shows the ratio of sgNT L + P treated to sgATG7 L + P treated lysoIP samples. PKA signaling complex highlighted in orange circles, along with canonical autophagic receptors. (C) Immunofluorescence of HEK293T sgNT and sgATG7 that over-express Flag-PKA RIα were treated with 500 nM BafA1 and 250 nM Torin for 1 h before fixation and immunostaining for Flag and LAMP2. 10 µm scale bar. (D) Quantification of Flag and LAMP2 co-localization from 8 non-overlapping fields, with at least 15 cells per field. ****$P$(adj) = 3.48919$E$ - 05, unpaired *t* test. (E) Immunofluorescence of HEK293T endogenously expressing mNeonGreen-PKA Cα in cells treated with shLuciferase control or shFIP200. Cells were serum-starved O/N then treated with 500 nM BafA1 or DMSO for 5 h before fixing and immunostaining and imaging for mNeonGreen and LAMP2. 10 µm scale bar. (F) Quantification of the co-localization of mNeonGreen-PKA Cα with LAMP2 under all conditions for 7 non-overlapping fields, with at least 10 cells per frame. ***$P$(adj)=0.0002, unpaired *t* test. Source data are available online for this figure.

## Phosphoproteomic analysis in iNeurons reveals AKAP11-dependent PKA regulation

The ability of AKAP11 to both scaffold PKA holoenzymes to the autophagosome, and to promote their autophagy-dependent degradation, prompted us to dissect its regulatory roles toward PKA signaling. Because AKAP11 is highly expressed in the brain, and given the strong association between AKAP11 gene truncations and SZ/BP (Herzog et al, 2023; Liu et al, 2023; Palmer et al, 2022), we carried out these experiments in induced pluripotent stem cell (iPSC)-derived neurons. Using CRISPR-interference (CRISPRi), we first suppressed AKAP11 expression in iPSCs that can be differentiated into inducible cortical neurons (i3 neurons) via doxycycline-driven expression of neurogenin-2 (NGN2) (Fernandopulle et al, 2018). Three independent single-guide RNAs (sgRNAs) effectively silenced AKAP11 both prior to differentiation (Fig. EV4A) and at 7 days post-NGN2 induction (Fig. 5A). Consistent with the results in HEK-293T cells, all AKAP11-targeting sgRNAs led to strong stabilization of RIα and, to a lesser extent, of Cα in both iPSCs (Fig. EV4A) and i3 neurons (Figs. 5A and EV4B–D). AKAP11-depleted i3 neurons were morphologically similar to control neurons (Fig. 5B).

We employed phosphoproteomic analysis to determine the effect of AKAP11 loss on PKA-dependent signaling in i3 neurons (Fig. 5C). Because RIα and Cα stabilization were clearly detectable under standard growth conditions (Figs. 5A and EV4B–D), we carried out our analysis without further stimulating PKA activity or autophagy. Of 7684 total identified and quantified phosphopeptides, 228 phosphopeptides (belonging to 174 proteins) were significantly altered (increased or decreased phosphorylation) in at least 2 out of 3 sgAKAP11 i3 neuron lines (Fig. 5D–F; Dataset EV2: phospho-peptide signals normalized by total protein levels). Notably, only 5.7% (13 sites) of differentially phosphorylated peptides matched the consensus PKA target site, whereas 94.3% were not predicted to be PKA sites (Dataset EV2).

Pathway enrichment analysis of the 174 differentially phosphorylated proteins revealed that, in line with its reported functions in non-neuronal lines (London et al, 2020). AKAP11 controls phosphorylation of proteins involved in microtubule polymerization/depolymerization and regulation of the actin cytoskeleton (Logue et al, 2011; Logue et al, 2011; Whiting et al, 2016) (Fig. 5G,H; Dataset EV2). Other differentially phosphorylated categories included synaptic function, neuronal morphogenesis and signaling (Fig. 5G,H; Dataset EV2).

Of the 13 bona fide PKA target sites differentially regulated in an AKAP11-dependent manner, 7 sites were hypophosphorylated in AKAP11-depleted neurons, whereas 6 sites were hyperphosphorylated. A notable site was Ser12 on the large ribosomal protein 34

(RPL34), which was among the most hypophosphorylated sites in all three AKAP11-depleted lines (Fig. 5D–F; Dataset EV2). Ser12 is highly conserved, is located in the extended N-terminal region of RPL34, and lies in close proximity to several ribosomal RNA (rRNA) molecules within the large subunit core (Anger et al, 2013) (Fig. EV4E,F). The position of Ser12 deep within the large ribosomal subunit core predicts that its PKA- and AKAP11-dependent phosphorylation could significantly impact ribosomal large subunit assembly and stability. According to published phosphoproteomic datasets (www.phosphosite.org), Ser12 is the most commonly detected phospho-site on RPL34, supporting its role as a putative regulatory site for the ribosome.

Among the most hypophosphorylated non-PKA substrate peptides in all three AKAP11-deleted i3 neuron lines was Ser83 in the RIα subunit (Fig. 5D–F; Dataset EV2). Ser83 is highly conserved, is located within the linker–hinge region of RIα, N-terminally proximal to the inhibitory sequence (IS) (Fig. EV5A,B), and phosphorylation of several residues in this region had previously been proposed to regulate the stability of IS binding to the catalytic cleft of Cα and, thus, the efficiency of Cα inhibition (Han et al, 2013; Haushalter et al, 2018). Ser83 is a commonly detected phospho-site on RIα (Boeshans et al, 1999; Gupte et al, 2006; Han et al, 2013) (www.phosphosite.org), thus it is likely to represent a bona fide regulatory site that, upon phosphorylation by upstream kinases (see "Discussion") may fine-tune the activity levels of AKAP11-bound PKA.

Finally, in addition to RIα, its brain-specific paralog RIβ, and Cα levels, total proteomic analysis revealed significant changes in the levels of numerous other proteins. Protein classes that were especially increased in AKAP11-depleted i3 neurons included ribosome biogenesis and rRNA processing factors, an effect possibly linked to our finding of AKAP11- and PKA-dependent RPL34 phosphorylation on Ser12 (Figs. EV4B–D and EV5C,D; Dataset EV2).

In summary, the phosphoproteomic analysis in i3 Neurons reveals a complex network downstream of AKAP11, whereby both PKA-dependent and PKA-independent phosphorylation events are either promoted or antagonized in an AKAP11-dependent manner.

## AKAP11-dependent RIα phosphorylation at Ser83 modulates PKA activation

The phosphoproteomic results in AKAP11-depleted i3 neurons are consistent with two possible roles for the interaction of the AKAP11-RIα- Cα complex with the autophagic machinery. One is a scaffolding function that facilitates PKA-dependent phosphorylation of a subset of downstream substrates, as well as regulation of PKA itself by upstream regulators; the second is an inhibitory role,

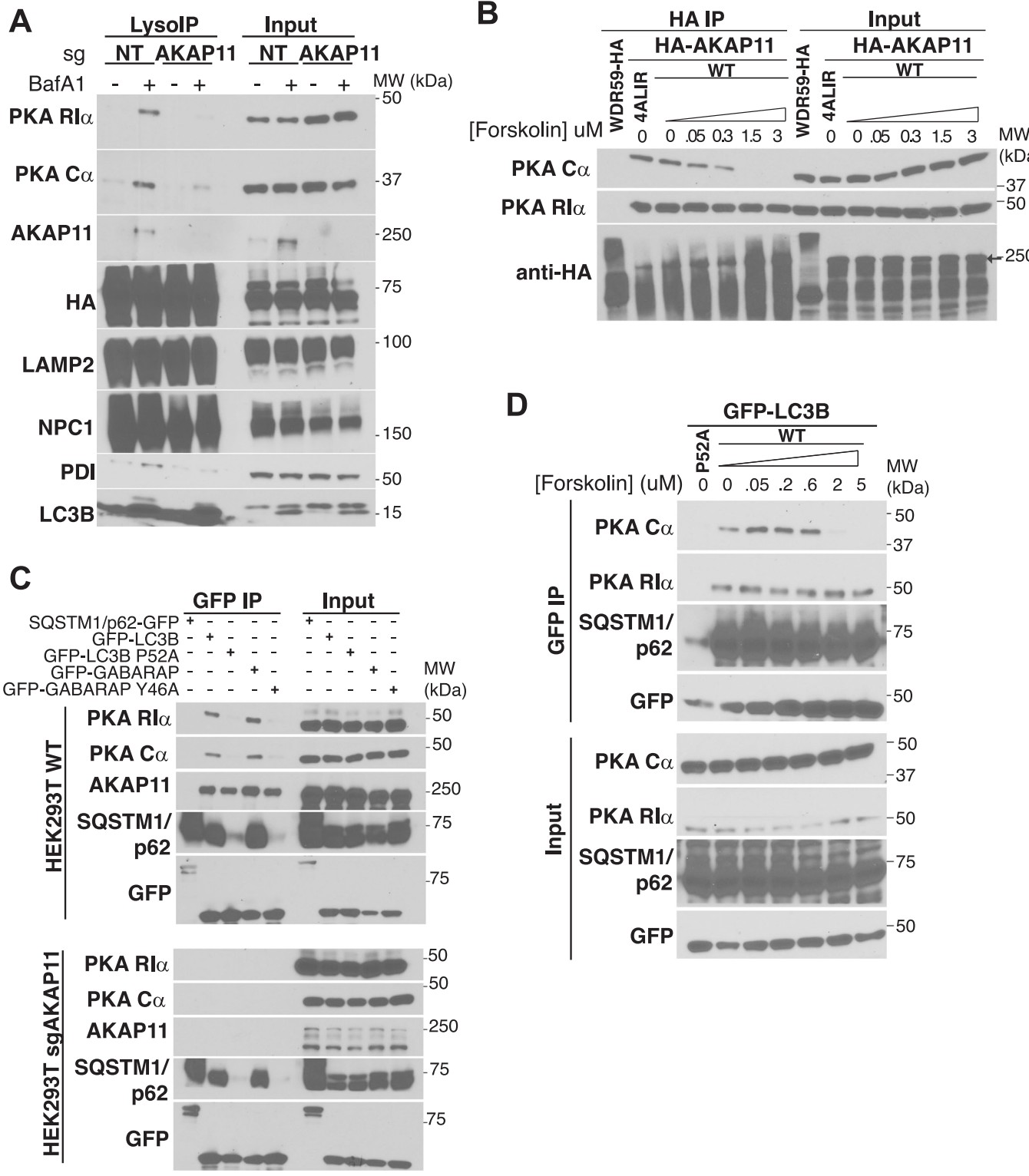

**Figure 3.  The PKA holoenzyme is degraded via AKAP11.**

(A) Immunoblots of LysoIP samples from HEK293T sgNT or sgAKAP11 cells, treated with 500 nM BafA1 or DMSO for 5 h. In the AKAP11 blot, asterisk indicates a nonspecific band. (B) Immunoblots from HEK293T transiently transfected with the WDR59-HA or with the indicated HA-AKAP11 cDNA constructs (the 4ALIR mutant is described in Fig. 4). Cells were treated with increasing concentrations of Forskolin, as indicated, for 20 min in complete media (10% FBS DMEM). In the HA blot, arrow indicates the position of the HA-AKAP11 protein. (C) Immunoblots of GFP immunoprecipitates from HEK293T sgNT or sgAKAP11 cells stably expressing the indicated GFP-tagged protein. (D) Immunoblots from HEK293T sgNT and sgAKAP11 cells stably expressing GFP-LC3 WT or P52A mutant. Cells were treated with increasing concentrations of Forskolin, as indicated, for 20 min in complete media (10% FBS DMEM) before lysis and GFP immunoprecipitation. Source data are available online for this figure.

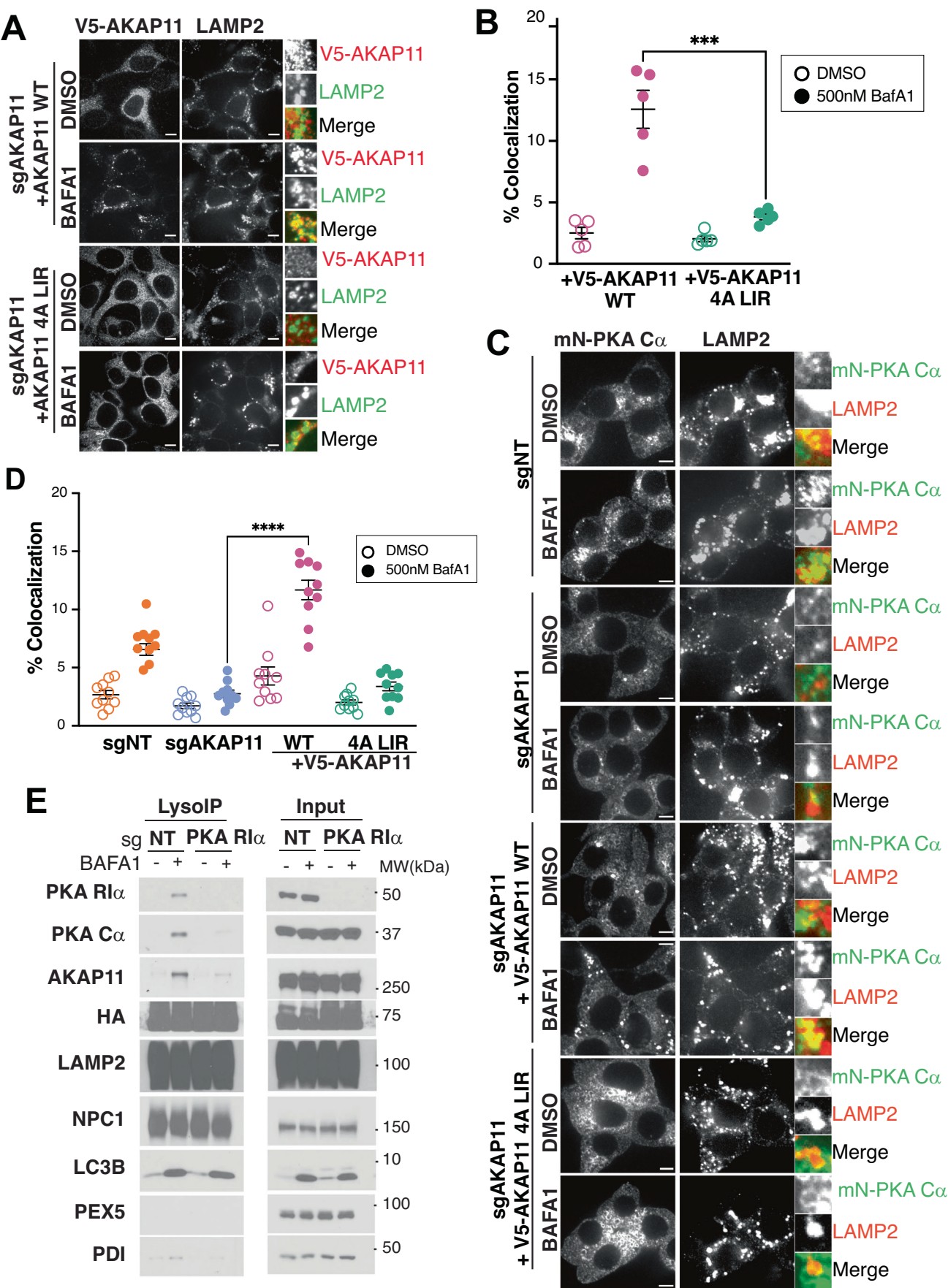

**Figure 4. The AKAP11 LiR motif mediates autophagic capture of the PKA holoenzyme.**

(A) Immunofluorescence from HEK293T sgAKAP11 cells stably expressing V5-AKAP11 WT or V5-AKAP11 4ALIR (WSNL > AAAA) treated with 500 nM BafA1 or DMSO for 5 h before fixing and immunostaining for V5 and LAMP2. 10 μm scale bar. (B) Quantification of V5 and LAMP2 co-localization from 6 non-overlapping fields, with at least 3 cells per field; ***P(adj.)=0.0005, unpaired t test. (C) Immunofluorescence of HEK293T sgNT or sgAKAP11 with endogenously-tagged mNeonGreen-PKA Cα also stably expressing V5-AKAP11 construct as indicated. Cells were serum-starved O/N then treated with 500 nM BafA1 or DMSO for 5 h before immunostaining with LAMP2. 10 μm scale bar. (D) Quantification of mNeonGreen and LAMP2 co-localization from 10 non-overlapping fields, with at least 3 cells per field. ****P(adj)= 1.00112E-08, unpaired t test. (E) Immunoblots of LysoIP samples from HEK293T sgNT or sgPKA RIα, treated with 500 nM BafA1 or DMSO for 5 h. Source data are available online for this figure.

in which autophagy-dependent degradation of the PKA holoenzyme suppresses phosphorylation of other PKA substrates.

To dissect how these activating and inhibitory functions relate to one another, we carried out parallel whole-cell phosphoproteomic analysis in control, AKAP11-deleted and AKAP11-deleted cells reconstituted with exogenous wild-type AKAP11 or with the AKAP11[4ALIR] mutant (Fig. 6A). We employed HEK-293T cells for these experiments as they were more amenable than the i3 neuron system to deletion-reconstitution protocols. As expected, re-expressing (V5-tagged) wild-type AKAP11 in sgAKAP11 cells decreased the elevated endogenous RIα levels, whereas V5-AKAP11[4ALIR] failed to bring RIα down to wild-type levels, albeit not completely (Fig. EV6A).

By comparative phosphoproteomic analysis of the AKAP11-deleted and reconstituted samples (carried out, as in i3 neurons, under standard growth conditions), we detected 81 phosphorylated peptides that changed significantly as a function of AKAP11 status upon normalization by total levels of their respective proteins (Dataset EV3). We focused on phospho-peptides that, if increased by deletion of AKAP11, were decreased back toward control levels by re-expressing wild-type AKAP11 and, vice-versa, phospho-peptides whose abundance was decreased by AKAP11 deletion and restored by re-expressing wild-type AKAP11 (Fig. 6B,C; Dataset EV3).

Compared to i3 neurons, fewer canonical PKA substrates were differentially phosphorylated as a function of AKAP11 status in HEK-293T cells, possibly reflecting a predominant role for AKAP11 in the brain. However, in agreement with the i3 neurons, one of the most hypophosphorylated peptides in AKAP11-deleted cells that were restored by AKAP11[WT] re-expression was Ser83 in RIα (Fig. 6B,C). As mentioned above, the location of S83 within the hinge-loop region of RIα suggests that its phosphorylation may regulate Cα activation. Supporting this idea, phosphorylation of nearby Ser101 by cGMP-dependent protein kinase (PKG) was shown to destabilize IS-Cα interaction, thereby contributing to PKA activation (Haushalter et al, 2018).

To dissect the role of S83 phosphorylation, we carried out a modification of previously described semi-reconstituted phosphorylation assays (Haushalter et al, 2018; Novero et al, 2024). A fluorescent synthetic peptide encoding a PKA consensus phosphorylation sequence (LRRASLG) was incubated with cell lysates from RIα-deleted HEK-293T cells that were reconstituted with wild-type, phosphomimetic (S83E) or phospho-null (S83A) RIα constructs (Fig. EV6B). Adding the synthetic PKA substrate peptide to the RIα-deleted lysates resulted in partial, cAMP-stimulated peptide phosphorylation by endogenous Cα present in the lysate (Fig. 6D,E).

In lysates from cells reconstituted with wild-type RIα, reformation of the PKA holoenzyme partially inhibited Cα activity, leading to significant reduction of the phosphorylated peptide upon cAMP addition (Fig. 6D,E). The phospho-null RIα S83A mutant was less capable of inhibiting cAMP-induced Cα kinase activity, resulting in more phosphorylated peptide. Conversely, the phospho-mimetic RIα S83E mutant inhibited substrate peptide phosphorylation more potently than wild-type RIα (Fig. 6D,E). Consistent with a previous report (Haushalter et al, 2018) mutating Ser 101, the target of PKG, to Ala also impaired inhibition of Cα kinase activity by RIα.

To determine whether the different peptide phosphorylation efficiencies reflect different strength of association between RIα and Cα, we carried out co-immunoprecipitation experiments between FLAG-tagged, wild-type or mutant RIα, and endogenous Cα (Fig. 6F). These experiments showed that, relative to wild-type RIα, the phosphomimetic RIα S83E mutant had increased binding to Cα in cAMP-stimulated conditions, providing a rationale for the increased potency of this mutant at inhibiting Cα (Fig. 6F). In contrast, we did not detect destabilization of the RIα−Cα interaction by the RIα S83A mutation, likely reflecting a more subtle effect of this mutation.

Because, like in i3 neurons, RIα phosphorylation at Ser83 was strongly decreased in AKAP11-depleted cells, AKAP11 may favor interaction of the bound PKA holoenzyme with one or more S83-phosphorylating upstream kinase(s) (see Discussion). Notably, Ser83 phosphorylation was significantly increased in AKAP11-deleted cells reconstituted with AKAP11[4ALIR] mutant compared to AKAP11[WT]-reconstituted cells, even after normalizing for the total levels of RIα protein (Fig. 6G). Thus, scaffolding of the Cα−RIα-AKAP11 complex on autophagosomes may restrict its accessibility by S83-phosphorylating kinases, thereby promoting PKA activation.

The phosphoproteomic analysis in this reconstituted system identified additional AKAP11-dependent signaling events that were regulated by AKAP11 interaction with the autophagic machinery. AKAP11 deletion caused increased phosphorylation of NCK associated protein 5 like (NCKAP5L), a microtubule plus ends-binding protein that regulates microtubule bundling and acetylation (Mori et al, 2015) at two non-PKA sites: Ser440 and Ser767. Both phosphorylation events were suppressed back to control levels by re-expressing wild-type AKAP11 but, notably, not by AKAP11[4ALIR] (Fig. 6B,C,G). Thus, the AKAP11 interaction with LC3B and GABARAP is required for AKAP11 to promote NCKAP5L dephosphorylation, further supporting that association of the PKA holoenzyme with autophagosomes modulates AKAP11-dependent signaling.

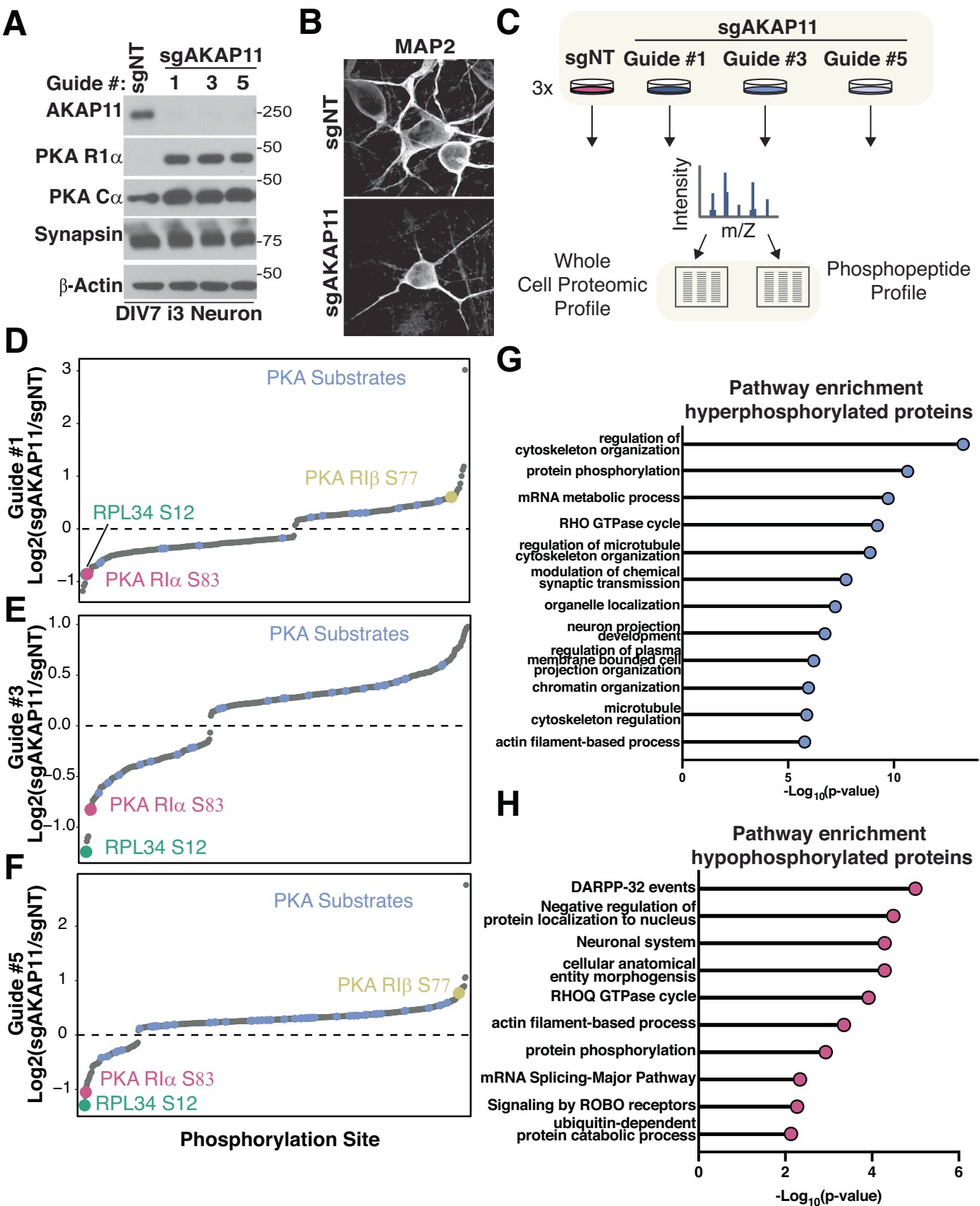

**Figure 5.   AKAP11 regulates PKA-dependent and independent signaling in i3 Neurons.**

(A) Immunoblot of sgNT- or sgAKAP11-treated i3 Neurons (DIV7) showing degree of AKAP11 depletion by the indicated guide. (B) Immunofluorescence images of sgNT or sgAKAP11 i3 Neurons (DIV7) immunostained with the neuronal dendritic marker, MAP2. (C) Schematic depicting mass spectroscopy workflow. Mass spectroscopy analysis of phosphorylated peptides was performed comparing i3 Neurons transduced with three sgRNA guides (#1, 3, 5) against AKAP11 vs a non-targeting guide. (D–F) After normalizing each phosphorylated peptide to the total protein abundance in the sample, waterfall plots of the foldchanges were generated (cutoff: $P$ value < 0.05) comparing each of the three AKAP11 guides to the non-targeting guide. (G, H) Bar graph depicting pathway enrichment analysis of phosphoproteomic dataset. (G) Pathway enrichment analysis of phosphoproteins enriched in sgAKAP11 i3 Neurons. Phosphorylated proteins with $Log_2$Foldchange ≥0.2 and significant $P$ value (<0.05), two-tailed unpaired $t$ test, were considered for pathway enrichment analysis. (H) Pathway enrichment analysis of phosphoproteins depleted in sgAKAP11 i3 Neurons. Phosphorylated proteins with $Log_2$Foldchange of ≤0.2 and significant $P$ value (<0.05), two-tailed unpaired $t$ test, were included for pathway enrichment analysis. Source data are available online for this figure.

Collectively, these data point to a model where binding of AKAP11 to autophagosomes regulates its bound PKA holoenzyme in multiple ways: (1) by gating access to PKA by upstream RIα-phosphorylating kinases that may fine-tune the effect of local cAMP, (2) by regulating signaling processes downstream of AKAP11 (potentially both PKA-dependent and PKA-independent), (3) by promoting autophagic degradation of the entire complex. All these processes contribute to the correct phosphorylation of both PKA-dependent and independent sites, enabling precise regulation of a broad signaling program that may play important roles in neuronal cell homeostasis (Fig. 7).

## Discussion

We undertook this study with the initial goal of identifying multiprotein signaling complexes that associate with, and may be regulated by the autophagy machinery. Lysosomal immunoisolation coupled with proteomic analysis identified the AKAP11-bound PKA holoenzyme as a prominent kinase complex that is scaffolded to the autophagosome and is targeted for autophagic degradation. Through follow-up functional experiments and phosphoproteomic analysis, we uncover new molecular functions of AKAP11, provide insights into how its autophagic capture modulates PKA signaling, and begin to shed light on how loss of AKAP11, an event strongly linked to the pathogenesis of SCZ/BP, impacts phosphorylation events inside neuronal cells. Our study thus uncovers a critical role for autophagosomes as physical scaffolds that govern AKAP11-PKA access to upstream regulators and downstream substrates, as well as control the overall levels of this complex.

One expected function of AKAP11 is to bring the PKA holoenzyme in proximity to specific substrate proteins, facilitating their phosphorylation upon local elevation of cAMP levels and de-inhibition of bound Cα. Our phosphoproteomic results indeed support this role for AKAP11 (Logue et al, 2011; Whiting et al, 2015). However, they also identify two additional functions of AKAP11. One is its ability to cause degradation of its bound RIα and Cα in autophagosomes; the second is to enable regulatory phosphorylation of its bound RIα by one or more upstream kinases, leading to stronger inhibition of Cα. The scaffolding function predicts that ablation of AKAP11 should lead to decreased phosphorylation of a subset of PKA substrates, whereas the pro-autophagic and RIα-regulating functions predict increased phosphorylation of certain PKA substrates upon AKAP11 loss (Fig. 7).

Indeed, phosphoproteomic experiments in i3 neurons show that AKAP11 loss increased phosphorylation of some canonical PKA substrates, while decreasing phosphorylation of others (Fig. 5D;

Dataset EV2). Among the most hypophosphorylated PKA substrates upon AKAP11 depletion was Ser12 of RPL34; we speculate that AKAP11 may bring the RIα- Cα in proximity to RPL34 at some point during the life cycle of the ribosomal large subunit, thus favoring Ser12 phosphorylation.

Conversely, in AKAP11-defective cells the total levels of RIα and, to a lesser degree, Cα were increased, whereas the inhibitory phosphorylation of Ser83 on RIα was decreased. Combined, these effects may result in association of more numerous, overactive holoenzyme molecules with other AKAP proteins, leading to hyperphosphorylation of other PKA substrates. Indeed, depleting AKAP11 from i3 neurons led to increased phosphorylation of canonical PKA sites on multiple proteins. Thus, AKAP11 emerges as a modulator that, via its scaffolding and autophagic adaptor functions, shapes both the strength and specificity of PKA signaling toward distinct classes of substrates.

Ser83 on RIα scored as the top AKAP11-dependent phosphorylated site. Although Ser83 is not in the IS segment, it lies nearby in a flexible loop that is not resolved in the numerous published structures of RIα but can be visualized in Alphafold predictions (Fig. EV5A). Though the exact molecular details of how Ser83 phosphorylation alters PKA activity is difficult to determine without structural information, the Alphafold model suggests that a phospho-group on Ser83 side chain may alter the packing of this flexible loop against PKA Cα, possibly forming a salt bridge with nearby Arg257 and thus stabilizing the interaction with the catalytic subunit.

How AKAP11 promotes RIα phosphorylation on Ser83 remains to be determined. Based on high-throughput profiling of human S/T kinase specificity (Johnson et al, 2023), Ser83 is a target site for Kinase Interacting with Stathmin (KIS), several Tyrosine kinase-like (TKL) including Activin receptor type-1B (ACVR1B) and Bone morphogenetic protein receptor type-2 (BMPR2), as well as the Casein Kinase (CK) group including CKIδ and CKIγ2. Conceivably, AKAP11 may bind to one or more of these kinases, holding them in physical proximity to the RIα linker-hinge region. Interestingly, single nucleotide polymorphisms in the KIS gene have also been associated with schizophrenia (Puri et al, 2007, 2008).

The observation that, in cells expressing the AKAP11[4ALIR] mutant, Ser83 phosphorylation was increased after normalizing for total RIα levels suggests that scaffolding of the Cα-RIα-AKAP11 complex to autophagosomes may regulate access of Ser83-phosphorylating kinases to RIα. The progressive maturation of an open phagophore into a closed autophagosome may gradually restrict access of LC3-bound PKA to certain upstream regulators and downstream substrates, while favoring others (Melia et al,

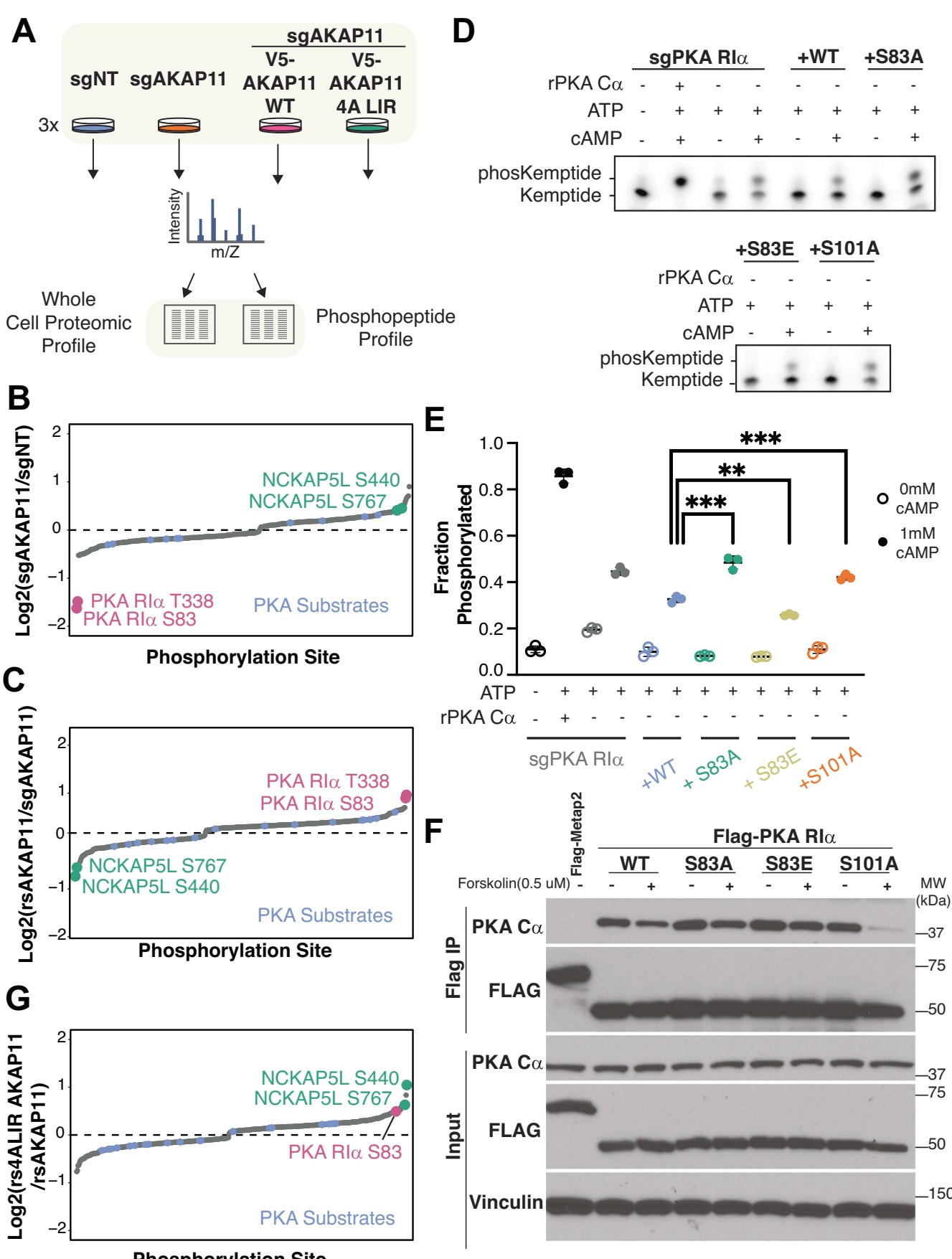

**Figure 6.    AKAP11 Regulates phosphorylation of RIα to modulate PKA activity.**

(A) Schematic depicting mass spectroscopy workflow. (B, C) Mass spectroscopy analysis of phosphorylated peptides was performed comparing AKAP11 genotypes. After normalizing each phosphorylated peptide to the total protein abundance in the sample, waterfall plots were generated of the foldchanges with statistical significance (*P* value < 0.05) comparing (B) sgAKAP11 knockout cells to control sgNT cell and (C) rescue V5-AKAP11 WT cells to sgAKAP11 knockout cells. In each waterfall plot, RIα phosphorylation sites have been annotated along with other sites of significance. (D, E) Analysis of in vitro PKA holoenzyme activity using adapted KiMSA activity assay. Fluorescent peptide shift comparing inhibitory capacity of Flag-R1α WT, phosphomimetic mutant (S83E) or phospho-null mutant (S83A), and S101A from HEK293T-sgR1α lysates. 0.2uM recombinant Cα, 2 mM ATP and 1 mM cAMP added where indicated. (E) In gel fluorescence quantification of *n* = 3 technical triplicates of in vitro PKA kinase activity. Error bars presented as mean ± SEM, **P = 0.0014, ***P = 0.0009, unpaired *t* test. (F) Immunoblot of co-immunoprecipitation from HEK293T-sgRIα that were transiently transfected with Flag-Metap2 or Flag-RIα constructs with indicated mutations. Cells were serum starved (0.5% FBS + DMEM) for 2 h, then stimulated with 0.5 µM forskolin or DMSO for 20 min. (G) Waterfall plot of statistically significant foldchanges between sgAKAP11 cells rescued with 4ALIR AKAP11 vs WT AKAP11 rescued sgAKAP11 cells. Source data are available online for this figure.

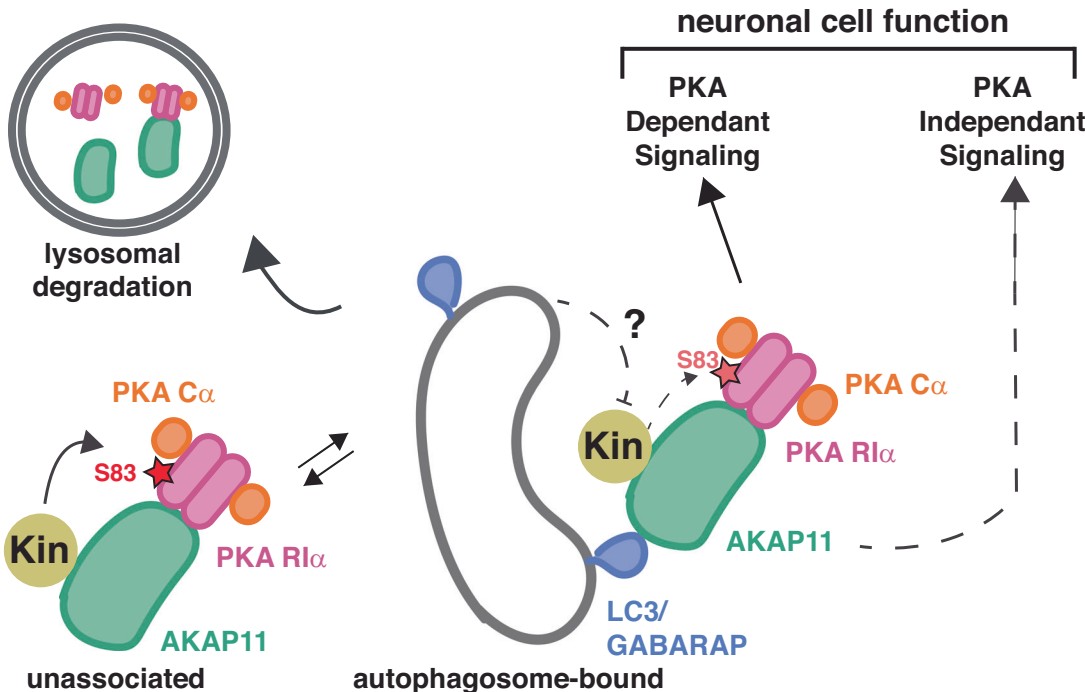

**Figure 7.    Model for AKAP11-dependent regulation of PKA signaling.**

The interaction of AKAP11 to LC3/GABARAP on the autophagosome is proposed to regulate association of AKAP11-bound PKA with upstream regulators, including the Ser83-phosphorylating kinase on RIα (labeled as 'Kin'). Subsequent closure of the autophagosome and fusion with lysosomes promotes wholesale degradation of the AKAP11-bound PKA. Blocking PKA-autophagosome association may lead to dysregulated PKA signaling, via both increased RIα phosphorylation of S83 and wholesale stabilization of the holocomplex.

2020; Yamamoto et al, 2023). However, our data are also consistent with a model in which the autophagy-promoting function of AKAP11 is independent from its role as a scaffold for the Ser83-phosphorylating kinase. Further studies are required to distinguish between these alternative models.

Our data do not support a model where selective autophagic degradation of RIα frees up Cα in order to amplify PKA signaling (Deng et al, 2021; Overhoff et al, 2022). First, under all paradigms tested we found that a pool of cellular Cα was scaffolded to ATG8 proteins via RIα and AKAP11, and was degraded along with RIα in an autophagy- and AKAP11-dependent manner, the only exception

being when we induced cAMP levels that are considered supra-physiological. This finding is in line with reports that the R- and C-subunits remain associated, at least in part, under conditions where PKA is catalytically active (Smith et al, 2013, 2017). Second, ablation of AKAP11 in i3 neurons and HEK-293T cells did not uniformly lead to suppression of PKA signaling, instead it resulted in some substrates being hyperphosphorylated, while others were hypophosphorylated.

These results indicate that AKAP11-dependent autophagic degradation of PKA is not a general 'brake-release' mechanism for PKA signaling. Instead, it could represent a fine-tuning

mechanism that reduces the amount of PKA that is allowed to get into contact with specific substrates, as well as the duration of these PKA-substrate interactions. It should be noted that AKAP11-associated PKA activity has not been linked to specific G-protein coupled receptors (GPCRs). The localization of AKAP11-PKA in intracellular vesicles such as autophagosomes may not allow its canonical regulation through rapid cycles of local cAMP production by adenylyl cyclase followed by cAMP breakdown by phosphodiesterases (Lefkimmiatis and Zaccolo, 2014; Truong et al, 2021). Instead, proper regulation of this complex may rely on wholesale degradation via autophagy, coupled with phosphorylation in critical sites such as the hinge-loop region of RIα, which could reinforce or antagonize the effects of local cAMP concentrations.

Our phosphoproteomic analysis reveals that the signaling functions of AKAP11 go well beyond direct regulation of PKA. In the i3 neuron model, close to 95% of differentially phosphorylated peptides were non-PKA substrates. Some of these altered phosphorylation events may be indirectly linked to dysregulated PKA. However, given the ability of AKAP11 to scaffold other signaling proteins such as GSK3β and PP1, AKAP11-dependent but PKA-independent regulatory actions are likely and must be taken into account when considering how AKAP11 gene loss may alter neuronal cell homeostasis and circuitry, leading to SZ/BP.

# Methods

### Reagents and tools table

| Reagent/ resource | Reference or source | Identifier or catalog number |
|---|---|---|
| **Experimental models** | | |
| HEK293T sgNT | Abu-Remaileh et al, 2017 | |
| HEK293T sgATG7 | Abu-Remaileh et al, 2017 | |
| HEK293T FIP200−/− | An et al (2020) | |
| HEK293T sgNT (Hygro) | This study | |
| HEK293T sgAKAP11 | This study | |
| HEK293T sgPKA RIa | This study | |
| HEK293T sgAKAP11 | Deng et al (2021) | |
| HEK293T-mNeonGreen-PKA Ca | Cho et al (2022) | |
| CRISPRi- i3iPSCs -NGN2 inducible cassette- dCas9-BFP-KRAB | Tian et al (2019) | |
| CRISPRi- i3iPSCs -NGN2 inducible cassette- dCas9-BFP-KRAB- sgAKAP11 | This study | |
| **Recombinant DNA** | | |
| pLJM1-TMEM192-RFP-3xHA | Lim et al, 2019 | |
| pLKO.1-shLuciferase | The RNAi Consortium, Broad Institute | |
| pLKO.1-shFIP200 | The RNAi Consortium, Broad Institute | |
| pLJM1-V5-AKAP11 WT | This study | |

| Reagent/ resource | Reference or source | Identifier or catalog number |
|---|---|---|
| pLJM1-V5-AKAP11 4 A LIR | This study | |
| pLJM1-V5-AKAP11 DS1S2 | This study | |
| pCGN-HA-AKAP11 | Deng et al (2021) | |
| pCGN-HA-AKAP11 4ALIR | Deng et al (2021) | |
| pLVX-Flag-PKA RIa | This study | |
| pLJM1-Flag-GFP-LC3B | This study | |
| pLJM1-Flag-GFP-LC3B P52A | This study | |
| pLJM1-Flag-GFP-GABARAP | This study | |
| pLJM1-Flag-GFP-GABARAP Y46A | This study | |
| pLJM1-p62-GFP | This study | |
| **Antibodies** | | |
| Anti-AKAP11 | | LS-C374339-100 |
| Anti-Flag | | #14793 |
| Anti-HA | | #3724 |
| Anti-PKA RIa (D54D9) | | #5675 |
| Anti-PKA Ca (D38C6) | | #5842 |
| Anti-LC3B (D11) | | #3868 |
| Anti-Vinculin (E1E9V) | | #13901 |
| Anti-TAX1BP1 (D1D5) | | #5105 |
| Anti-V5 rabbit | | #13202 |
| Anti-V5 mouse | | #80076 |
| Anti- SQSTM1/p62 | | #397749 |
| Anti- PP2AA | | #2041 |
| Anti- HOP | | #5670 |
| Anti- eEF1A | | #2551 |
| Anti- CACYBP | | #3354 |
| Anti- HSP90 | | #4874 |
| Anti- CCT2 | | #3561 |
| Anti- PSMA2 | | #11864 |
| PP2AC | | #2259 |
| GPI | | #57893 |
| GSK3β | | #12456 |
| Synapsin | | #4297 |
| Oct4 | | #27505 |
| SQSTM1/p62 | | 18420-1-AP |
| eEF1D | | 10630-1-AP |
| CCT5 | | 67400-1-Ig |
| MAP2 | | 17490-1-AP |
| GFP | | SC-9996 |
| GFP | | A11122 |
| LAMP2 | | SC-18822 |
| PSMD7 | | ab11436 |
| PKAR2A | | VPA00905 |
| Tuj1 | | 801202 |
| Goat anti-Rabbit IgG, peroxidase labeled | | PI-1000; |

| Reagent/ resource | Reference or source | Identifier or catalog number |
| --- | --- | --- |
| | Goat anti-Mouse IgG, AlexaFluor568 labeled | #A11004 |
| | Goat anti-Rabbit IgG, AlexaFluor488 labeled | #A11011 |
| | Goat anti-Mouse IgG, AlexaFluor647 labeled | #A21235 |
| **Oligonucleotides and other sequence-based reagents** | | |
| **Chemicals, enzymes and other reagents** | | |
| | Bafilomycin A1 | J61835 |
| | Torin1 | Tocris, 4247 |
| | Leupeptin hemisulfate | J61188 |
| | Pepstatin A | 195368 |
| | Forskolin | 11018 |
| | GFP-trap | gta |
| | mNeonGreen-trap | nta |
| | Pierce Anti-HA magnetic beads | 88836 |
| | Anti-Flag beads | A2220 |
| | Lenti-X concentrator | #631232 |
| | Opti-prep | Sigma D1556 |
| | 5'FAM-Kemptide | Anaspec AS-29933-1 |
| | PhosStop | 4906845001 |
| | Normal Donkey Serum | 017-000-121 |
| **Software** | | |
| | BioGrid | https://thebiogrid.org/ |
| | Prism 10 | https://www.graphpad.com |
| | ImageJ/FIJI | NIH |
| | Cytoscape | https://cytoscape.org/ |
| | Metascape | https://metascape.org/ |
| | Gene ontology/Panther | https://geneontology.org/ |
| **Other** | | |

## Tools and reagents

Reagents were obtained from the following sources: antibodies to AKAP11 (LS-C374339-100) from LSBio. FLAG (#14793), HA (#3724) PKA RI-α (D54D9) (#5675), PKA C-α (D38C6) (#5842), LC3B (D11) (#3868), Vinculin (E1E9V) (#13901), TAX1BP1 (D1D5)(#13901), V5 rabbit (#13202), V5 mouse (#80076), SQSTM1/p62 (#397749), PP2AA (#2041), HOP (#5670), eEF1A (#2551), CACYBP (#3354), HSP90 (#4874), CCT2 (#3561), PSMA2 (#11864), PP2AC (#2259), GPI (#57893), GSK3β (#12456), Synapsin (#4297), Oct4 (#27505) from Cell Signaling Technologies. SQSTM1/p62 (18420-1-AP), eEF1D (10630-1-AP), CCT5 (67400-1-Ig), MAP2 (17490-1-AP) from Proteintech. GFP (SC-9996), LAMP2 (SC-18822) from Santa Cruz Biotechnology.) GFP

(A11122) from Invitrogen, PSMD7 (ab11436) from Abcam, PKAR2A (VPA00905) from BioRad, Tuj1 from Biolegend (801202). Bafilomycin A1 (Alfa Aesar, J61835), Torin1 (Tocris, 4247), Forskolin (Cayman Chemical: 11018), Lenti-X concentrator (Takara Bio #631232). GFP-trap (gta) and mNeonGreen-trap (nta) from Proteintech/Chromotek. Pierce Anti-HA magnetic beads (88836) from Thermo Scientific. Anti-Flag beads (A2220) from Sigma-Aldrich.

## Mammalian cell culture

HEK293T cells and their derivatives were cultured in DMEM base media with 10% (v/v) fetal bovine serum (FBS) and supplemented with 2 mM Glutamine, and 1% penicillin and streptomycin. All cell lines were cultured at 37 °C and 5% $CO_2$. All cell lines were free of mycoplasma contamination, routinely checked by mycoplasma PCR Detection Kit (abm, #G238) and/or DAPI staining.

## Lentivirus production and infection

Lentiviruses were made by co-transfecting pLJM1 constructs with psPAX2 and pMD2G packaging plasmids into HEK293T cells using PEI transfection reagent. Viral supernatant was collected after 48 h and again after 72 h post-transfection and filtered using 0.45 μm syringe filter. The collected virus was concentrated using Lenti-X concentrator (Takara Bio #631232) according to manufacturer's protocol and stored at −80 °C. For lentivirus transfection, target cells were seeded along with the virus and 10 μg/mL polybrene. After 24 h incubation, virus containing media was removed and fresh media containing puromycin (1 μg/mL) hygromycin (200 μg/mL), or blasticidin (5 μg/mL) was added for selection.

## Drug treatments

All drug treatments were performed as follows unless otherwise specified. Bafilomycin A1 (Alfa Aesar, J61835) was used at 500 nM for 5 h. Torin1 (Tocris, 4247) was used at 250 nM for 1 h. Forskolin (Cayman Chemical) was used at 1 μM for 25 min.

## Generation of CRISPR knockout cell lines

HEK293T sgATG7 were generated as described previously (Abu-Remaileh et al, 2017), sgFIP200 were generated as described previously (An et al, 2020).

To generate HEK293T sgAKAP11, sgPRKAR1A, the following targeting sequences were cloned into pLentiCRISPRv2 vector: AKAP11-5'-ATGTCCCAGGACATTCACTG-3', PRKAR1A- 5'-ACCAAAAGATTACAAGACAA-3'. Infected cells were selected for hygromycin resistance. Cells were maintained in selection medium for 3–5 days to ensure knockout. Knockouts were validated by immunoblotting and LysoIP.

## Generation of CRISPRi i3 neurons

To generate sgAKAP11 KD i3 Neurons, the following target sequences were cloned into pLG15 vector plasmid: sgNT 5'-GTCCACCCTTATCTAGGCTA-3',sgAKAP11[#1] 5'-AGCCTCCGCGGCGAGCACGT-3', sgAKAP11[#3] 5'-GGTGACATGTCTGTGAGCTG-3', sgAKAP11[#5] 5'-TCGGCGCCCGGCTCACCTGG-3'.

## i³Neuron differentiation

CRISPRi- i³iPSCs containing neurogenin2 (NGN2) inducible cassette and dCas9-BFP-KRAB were a generous gift from Dr. Michael Ward (Tian et al, 2019). CRISPRi- i³iPSCs were differentiated as previously described (Fernandopulle et al, 2018). Briefly, iPSCs were dissociated with Accustase (Gibco) and seeded onto matrigel (Corning) coated six-well plates. iPSCs were infected with lentiviruses concentrated in essential-8 medium (Gibco) expressing sgNT or sgAKAP11. After 24 h, media was replaced with fresh media containing puromycin (1 µg/mL, SUPPLIER). Media was replaced daily with fresh media containing puromycin for 72 h. After selection, cells were expanded for one passage before dissociating and seeding cells into induction medium containing N2-supplement (N2,Gibco) in Knockout DMEM/F:12 with non-essential amino acids, GlutaMax and doxycycline (NEAA, Gibco; GlutaMax,Gibco; Doxycycline 2 µg/mL,). Media was replaced with fresh induction media for 72 h, before dissociating cells and seeding onto PLO/Laminin coated plates (prepared as described in Fernandopulle et al, 2018) containing BrainPhys neuronal differentiation media supplemented with B27+ containing BDNF, NT-3, and GDNF. i³Neurons were given 60% media changes (remove 50%, add back 60%) every 3 days until harvested for experiments.

## Lysosome immunoprecipitation (LysoIP)

Lysosomes from cells lines stably expressing TMEM192-RFP-3xHA were purified as previously described (Lim et al, 2019). In brief, 18 million cells were seeded in a 15 cm dish, the following day cells were treated either with DMSO, 500 nM BAFA1 for 5 h or 20 µM Leupeptin + 20 µM Pepstatin for 24 h. All following steps were done with cold buffers or on ice to maintain samples at 4 °C. Cells were washed and then resuspended in 5 ml of PBS. Samples were spun at 320×g and pelleted cells were resuspended in 750uL K-PBS (136 mM KCl, 10 mM KH2PO4, pH 7.25, with addition of fresh 0.5 mM TCEP and Pierce Protease inhibitor tablet (Thermo A32965) with 3.6% (v/v) Opti-prep (Sigma D1556). Cells were mechanically lysed by five passages through a 23-gauge needle and post-nuclear fractions were collected after a 1370×g 10 min spin and incubated with Pierce Anti-HA magnetic beads (88836) beads for 20 min. Samples were washed 3 times with 1 mL of K-PBS using a magnetic stand. For western blot analysis, samples were eluted directly into 2×Urea sample loading buffer (150 mM Tris, pH 6.5, 6 M urea, 6% SDS (w/v), 25% (v/v) glycerol, 5% (v/v) β-mercaptoethanol, 0.02% (w/v) bromophenol blue) at room temperature overnight. For proteomics experiments, lysosomal immunoprecipitates were eluted from beads using 0.1% NP-40 in PBS for 30 min at 37 °C, beads were removed and the resulting eluate was snap-frozen with LN₂.

## LysoIP mass spectrometry and proteomic analysis

Samples were prepared in biological triplicates and analyzed by tandem mass spectrometry with an Orbitrap analyzer by the Proteomics Core Facility at the Whitehead Institute. The samples were TCA-precipitated, resuspended in TEAB, reduced and alkylated, digested, isotope-labelled, combined, cleaned up (SPE) and fractionated.

The resulting data was filtered to exclude contaminants listed in the cRAP database (common Repository of Adventitious Proteins, GPM) or in MaxQuant, and to include only proteins with peptides occurring in at least two replicates. Consequently, the dataset was normalized in Python: First, based on the mean total intensity per MS run, all MS channels were corrected for sample loading. Second, Internal Reference Scaling (IRS) was the last normalization step. Batchcorrection with Combat's PyCombat served as benchmark. For Principal Component Analysis, the data was standard-scaled. Following normalization, the foldchange between samples was calculated as the log base 2 of the ratio of the averages. Significance for each foldchange was determined by taking the -log base ten of a welch two-tailed *t* test.

## Bioinformatic analysis

List of lysosomal proteins was obtained experimentally (954 human proteins). Protein interactions between the initial list of lysosomal proteins and other interactors were determined based on BioGRID (https://thebiogrid.org/), a data base of protein–protein interactions from both high-throughput datasets and individual focused studies. The resulting list included the protein–protein interactome of the initial lysosomal protein list. The connectogram was constructed using Cytoscape (https://cytoscape.org/).

Gene ontology analysis was performed using Panther with default settings. The plot was generated in GraphPad Prism. Pathway enrichment analysis in i3 Neurons was performed using Metascape (Zhou et al, 2019). The plots were generated in GraphPad Prism.

## Immunofluorescence

Cells were seeded on fibronectin-coated glass coverslips in 6-well or 12-well plates the day before the experiment was to be performed. Cells were Bafilomycin or Torin treated where indicated and fixed using 4% (v/v) paraformaldehyde (PFA) for 15 min at RT. Cells were rinsed 3 times with PBS, permeabilized with 0.1% (w/v) saponin in PBS for 10 min at RT. Cells were rinsed in PBS 3 times. Primary antibodies were diluted in 5% (v/v) normal donkey serum (NDS) (Jackson ImmunoResearch) and incubated at RT for 1 h. The coverslips were rinsed with PBS 3 times. Coverslips were then incubated in fluorescently-conjugated secondary antibodies diluted to 1:4000 diluted in 5% (v/v) NDS for 45 min at RT while being protected from light. Coverslips were then rinsed with PBS 3 times and mounted on glass slides using VECTASHIELD Antifade Mounting Medium with or without DAPI (Vector labs).

## Microscopy

All confocal microscopy was using a spinning-disk confocal system built on a Nikon Eclipse Ti microscope (Nikon Instruments) with Andor Zyla-4.5 sCMOS camera system(Andor Technology) using a Plan Apo ×60 oil objective. Images of fine cellular detail were acquired with an additional ×1.5 magnifier.

## Image analysis

For quantification of co-localization, 5–10 non-overlapping images were acquired from each coverslip. Raw, unprocessed images were

imported into ImageJ v1.53 and converted to 8-bit images, and images of individual channels were thresholded independently to exclude background and non-specific staining noise and converted to binary masks. Co-localization analysis was assessed using the threshold 'AND' function. Percent co-localization was calculated by dividing co-localized value by lysosomal marker threshold.

## Cell lysis and co-immunoprecipitation

Cell lysates were prepared by removing media and rinsing cell monolayer once with DPBS and lysed in either of the following lysis buffers: Triton-based (1% Triton X-100 (v/v), 10 mM sodium β-glycerol phosphate, 10 mM sodium pyrophosphate, 4 mM EDTA, 40 mM HEPES, pH 7.4, and one EDTA-free protease inhibitor tablet per 50 ml) or CHAPS-based (0.3% CHAPS (w/v), 10 mM sodium β-glycerol phosphate, 10 mM sodium pyrophosphate, 2 mM EDTA, 20 mM HEPES, 150 mM NaCl, pH 7.6, one EDTA-free protease inhibitor tablet per 50 ml). Cells lysed on nutator for 10 min. Lysates were cleared via centrifugation using a microcentrifuge at $17,000 \times g$ for 10 min at 4 °C. Protein content in lysate samples were measured by Bradford assay or BCA. Samples of equal protein concentration and addition of 5X SDS sample buffer (235 mM Tris, pH 6.8, 10% (w/v) SDS, 25% (v/v) Glycerol 25% (v/v) β-mercaptoethanol, 0.1% (w/v) bromophenol blue) were prepared for SDS-PAGE.

For Flag or HA immunoprecipitations, cells were seeded in a 10 cm at a density appropriate for them to reach confluency after 24 h. Cells were lysed as described above. In total, 25 μl of a well-mixed 50% slurry of anti-Flag M2 affinity gel or HA magnetic beads were added to each lysate sample and incubated at 4 °C while on the rotator for 1 h. For GFP or mNeonGreen IP, 15 μl of well-mixed 50% slurry of GFP-Trap agarose or mNeonGreen-Trap agarose were added to each lysate sample and incubated at 4 °C while on the rotator for 1 h. Immunoprecipitant beads were washed three times with lysis buffer. Immunoprecipitated proteins were denatured by adding 100 μl of sample buffer, left overnight at RT or heating to 95 °C for 5 min.

## Immunoblotting

For immunoprecipitation, 10% of the total immunoprecipitated material was loaded per lane, and 0.5% of total input was loaded per lane. Proteins were transferred to a PVDF membrane (Millipore IPVH00010), blocked with 5% (w/v) non-fat milk in TBS-T, and incubated in primary antibodies (diluted in 5% milk in TBS-T) for 3 h at RT or overnight at 4 °C. Membranes were rinsed with TBS-T and incubated with horseradish peroxidase conjugated anti-rabbit or anti-mouse secondary antibodies (diluted in 5% milk in TBS-T) for 1 h at room temperature. Membranes were washed again with TBS-T and incubated with Pierce ECL Western Blotting Substrate (Thermo Scientific, 32109) before being exposed to Prometheus ProSignal Blotting Film (Genesee Scientific, 30-507L).

## Phosphoproteomic sample preparation

HEK293T sgNT or sgAKAP11 and sgAKAP11 stably expressing V5-AKAP11 WT or V5-AKAP11 dLIR were seeded ($5.0 \times 10^6$ cells) in 10 cm dish in DMEM complete medium (DMEM, 10% (v/v) FBS, 2 mM Glutamine, 1% (v/v) penicillin/streptomycin). After overnight adherence cells were switched to DMEM + 10% dFBS

medium. Cells were harvested 24 h later and lysed as previously described. Protein content in lysate samples were measured by BCA. Lysis was flash frozen in liquid nitrogen until mass spectrometry analysis.

## Phosphoproteomic mass spectrometry

All samples were labeled with iodoacetamide, and resuspended in 100 mM of 4-(2-Hydroxyethyl)-1-piperazinepropanesulfonic acid (EPPS) buffer, pH 8.5 and digested at 37 °C with trypsin overnight. The samples were labeled with TMT Pro and, quenched with hydroxylamine. All the samples were combined and desalted using a 100 mg Sep-Pak cartridge, followed by drying in a rotary evaporator. Phosphopeptides were enriched using a Fe-NTA spin column (Thermo Fisher). Flow through samples were dried, and fractionated with basic pH reversed phase (BPRP) high-performance liquid chromatography (HPLC) as described previously. Samples were desalted via StageTip and dried with speed vac. Samples were resuspended in 5% formic acid, and 5% acetonitrile for LC-MS/MS analysis. Mass spectrometry data were collected using Exploris 480 or Orbitrap Eclipse mass spectrometers (Thermo Fisher Scientific) coupled with a nLC-1200 or Vanquish Neo liquid chromatograph (Thermo Fisher Scientific), respectively, with a 90 or 150 min gradient across a Nano capillary column (100 μm D) packed with ~35 cm of Accucore C18 resin (Thermo Fisher Scientific). A FAIMSPro (Thermo Fisher Scientific) was utilized for field asymmetric waveform ion mobility spectrometry (FAIMS) ion separations. The instrument methods, which are embedded in the RAW files, included Orbitrap MS1 scans (resolution of 120,000; mass range 400–1600 $m/z$; automatic gain control (AGC) target $4 \times 10^5$, max injection time of 50 ms. MS2 scan parameters were set as described previously (CID collision energy 35%; AGC target $7.5 \times 10^3$; rapid scan mode; max injection time (50 mM)).

## Phosphoproteomics analysis

Comparative phospho-peptide analysis between genotypes was conducted by first normalizing each phospho-peptide to account for differences in total protein abundance in each genotype. As such, the relative abundance of each protein at the whole cell level in each genotype was compared to the levels in sgNT control samples then this normalized abundance was used to normalize the corresponding phosphopeptide for each replicate. Following normalization, the foldchange between samples was calculated as the log base 2 of the ratio of the averages. Significance for each foldchange was determined by performing a welch two-tailed $t$ test. Phosphopeptides fold changes with $P$ values < 0.05 were then plotted on a waterfall plot. All analysis was done using custom R scripts.

## Statistical analysis

All graphs were assembled, and statistics were performed using Prism 10 (GraphPad). Error bars on all graphs are shown as the mean ± SEM. The details of each statistical test performed are given in the legend accompanying each figure. Unless otherwise indicated, all co-localization analysis was performed on 10 non-overlapping fields that contain a minimum of 3 cells per field. Unless otherwise indicated, all proteomic measurements were performed on three independent biological replicates for each sample.

## Kinase activity assay

HEK293T-sgPKA RIα cells were seeded in 6 cm tissue culture-treated plates. After overnight incubation, cells were transiently transfected with 5 µg Flag-RIα WT type and mutant DNA using PEI, as described previously. Twenty-four hours post transfection, cells were lysed in the plate using 120 µl of cold NDP-40 Lysis buffer (1× PBS, 1% (v/v) NDP-40, 1× PhosStop, 1× EDTA-free protease inhibitor). Lysates were cleared via centrifugation using a microcentrifuge at 17,000×$g$ for 10 min at 4 °C and then normalized to 2 µg/µL. Each kinase reaction was made by using a previously described protocol with final concentrations of 200 mM Tris-HCl pH 7.4, 10 mM MgCl$_2$, 0.2 mM ATP, 0.5 mM TCEP, 1× EDTA-Free protease inhibitor, 1× PhosStop, 21 µM 5'FAM-Kemptide (Anaspec) and 0.4 µg/µL cell lysate. 1 mM cAMP was added where noted. Kinase reactions were incubated at RT for 20 min, followed by the addition of 5× SDS sample buffer (235 mM Tris, pH 6.8, 10% (w/v) SDS, 25% (v/v) Glycerol 25% (v/v) $\beta$-mercaptoethanol, 0.1% (w/v) bromophenol blue). In all, 2.5 µL of each sample were run on 12% NuPAGE Bis-Tris protein gels with MES running buffer. Blank lanes were left in between each sample lane and no protein ladder standard was used.

In-gel fluorescence was imaged using ChemiDoc. Fiji ("Fiji is just ImageJ") was used for quantification of fluorescence. Background subtraction was performed using a 50-pixel rolling ball subtraction followed by intensity measurements for each individual band. Fraction of substrate phosphorylated was calculated as phosphorylated intensity/(phosphorylated intensity + unphosphorylated intensity).

## Data availability

All data are available in the source data and appendix tables. This study includes no data deposited in external repositories.

The source data of this paper are collected in the following database record: biostudies:S-SCDT-10_1038-S44318-025-00436-x.

## Peer review information

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

## Acknowledgements

This work was supported by NIH 1R35GM149302, the Pew Innovation Fund and the Edward Mallinckrodt, Jr. Foundation Scholar Award to RZ, the National Science Foundation Graduate Research Fellowship to ASR. The authors thank R Irannejad for providing reagents and for critical reading of the manuscript. M Leonetti for sharing the mNeon-Cα HEK-293T cells. Z Yue for sharing AKAP11-deleted HEK-293T cells. H An and W Harper for the FIP200-deleted HEK-293T cells. M Ward for the CRISPRi-iPSCs containing neurogenin2 (NGN2) inducible cassette and dCas9-BFP-KRAB. We thank J Thorner and all members of the Zoncu lab for helpful discussion.

## Author contributions

**Ashley Segura-Roman**: Conceptualization; Data curation; Validation; Investigation; Visualization; Writing—original draft; Writing—review and editing. **Y Rose Citron**: Conceptualization; Data curation; Formal analysis; Validation; Investigation; Writing—original draft. **Myungsun Shin**: Formal analysis; Investigation. **Nicole Sindoni**: Investigation. **Alex Maya-Romero**: Investigation. **Simon Rapp**: Investigation. **Claire Goul**: Investigation. **Joseph D Mancias**: Supervision. **Roberto Zoncu**: Conceptualization; Supervision; Funding acquisition; Investigation; Writing—original draft; Project administration; Writing—review and editing.

Source data underlying figure panels in this paper may have individual authorship assigned. Where available, figure panel/source data authorship is listed in the following database record: biostudies:S-SCDT-10_1038-S44318-025-00436-x.

## Disclosure and competing interests statement

The authors declare no competing interests.

# Expanded View Figures

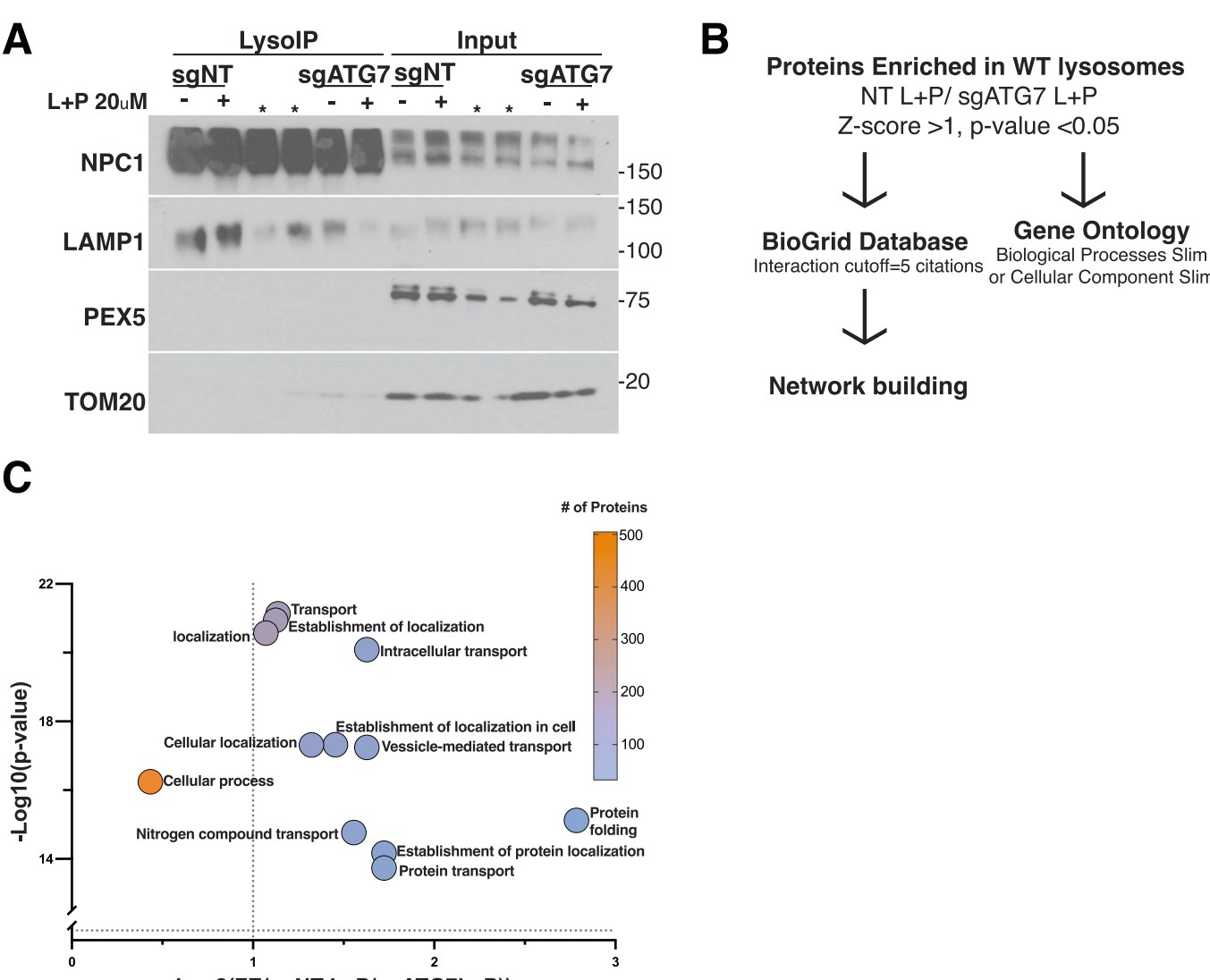

**Figure EV1. Lysosomal immunoisolation and network building.**

(A) Immunoblots of Lysosomal immunoprecipitation and corresponding input from HEK-293T-sgNT or sgATG7 after treatment 20 μM Leupeptin and 20 μM Pepstatin for 24 h to block lysosomal degradation of lysosomal substrates. (B) Outline depicting bioinformatic pipeline in which proteins identified as 'Hits' (Log2(FC[sgNT L + P/ sgATG7 L + P]) > 1, P value < 0.05, two-tailed t test) in LysoIP from WT and ATG7-null cells were subjected to custom-written pipeline where interactors of Hits (using citation cutoff of ≥5 citations) were identified using data from BioGrid. List of protein 'Hits' were also entered in panther to generate a gene ontology analysis for enrichment of biological processes or enrichment of cellular component. (C) Volcano plot of "biological processes slim" Go-terms of proteins enriched in wild-type lysosomes compared to autophagy-null lysosomes in 20 μM L + P 24-hour treatment. Protein list obtained from proteins identified as 'Hits'. P value calculated using Fishers exact test.

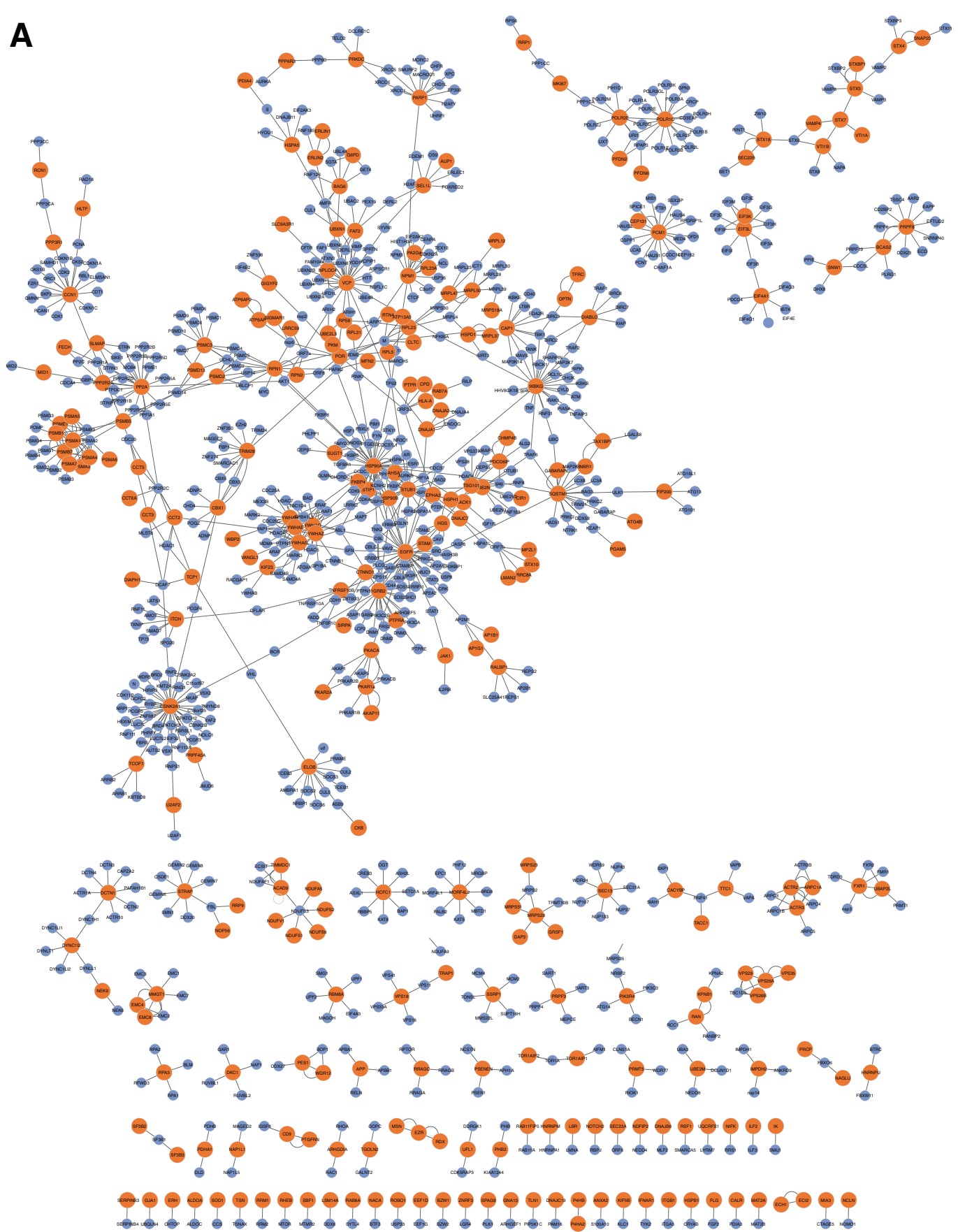

**Figure EV2.  Network of autophagic substrates.**

(A) Network representation of autophagy-dependent substrates (orange). Blue nodes show cited protein–protein interactions to provide context for which multi-protein complex substrates belong to.

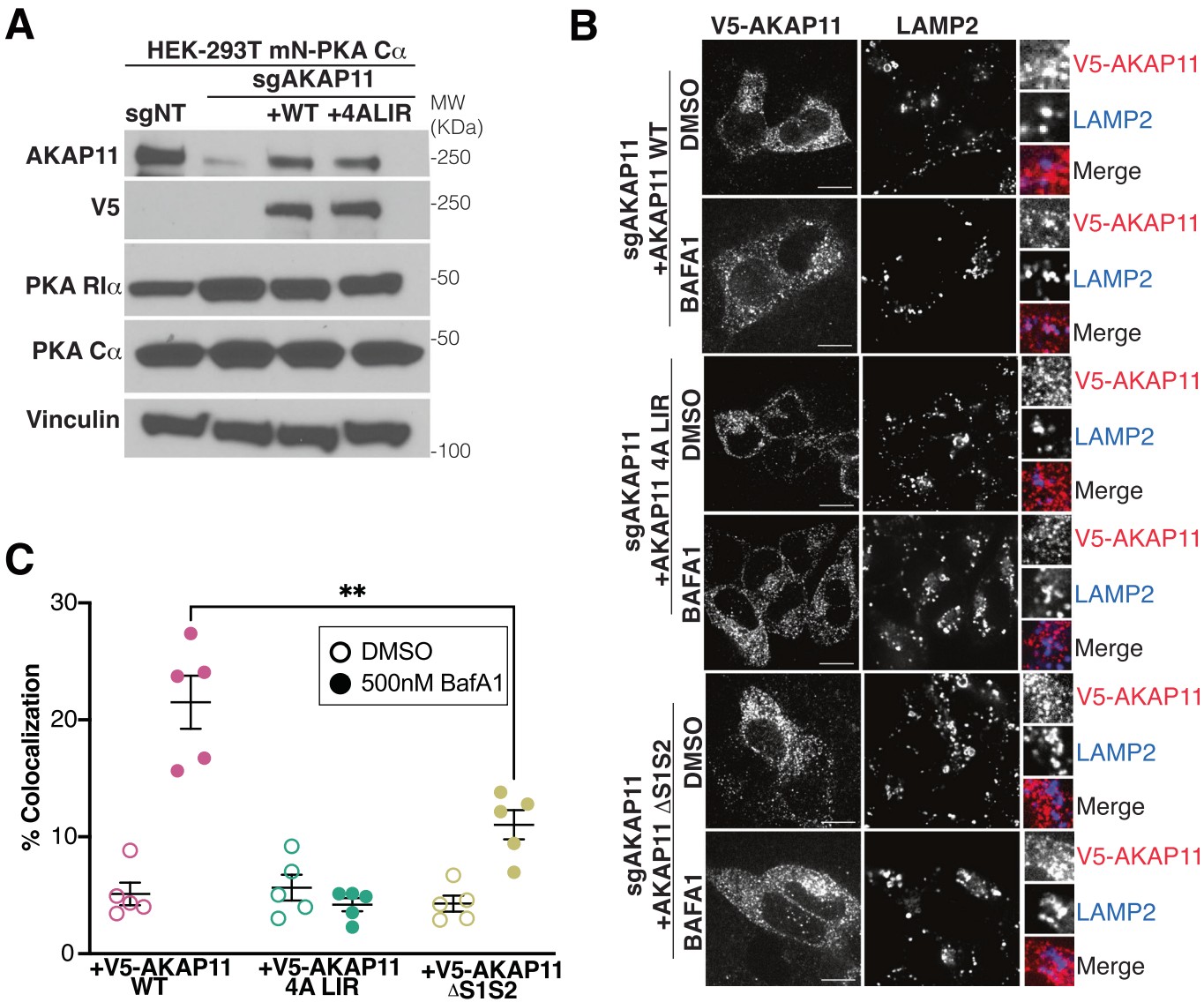

**Figure EV3.   PKA RIα binding defective AKAP11 has decreased lysosomal localization.**

(A) Immunoblot validation of AKAP11 knockout using whole cell lysate from HEK293T sgNT or sgAKAP11 and sgAKAP11 with stably expressing the indicated V5-AKAP11 construct. (B) Immunofluorescence from HEK293T sgAKAP11 cells stably expressing V5-AKAP11 WT, V5-AKAP11 4ALIR (WSNL > AAAA), or V5-AKAP11 ΔS1S2 (deleted AA 615–628 & 1650–1663) were treated with 500 nM BafA1 or DMSO for 5 h before fixing and immunostaining for V5 and LAMP2. 10 μm scale bar. (C) Quantification of V5 and LAMP2 co-localization from 5 non-overlapping fields, with at least 3 cells per field; error bars presented as mean ± SEM, **$P$(adj.)= 0.0037, unpaired $t$ test.

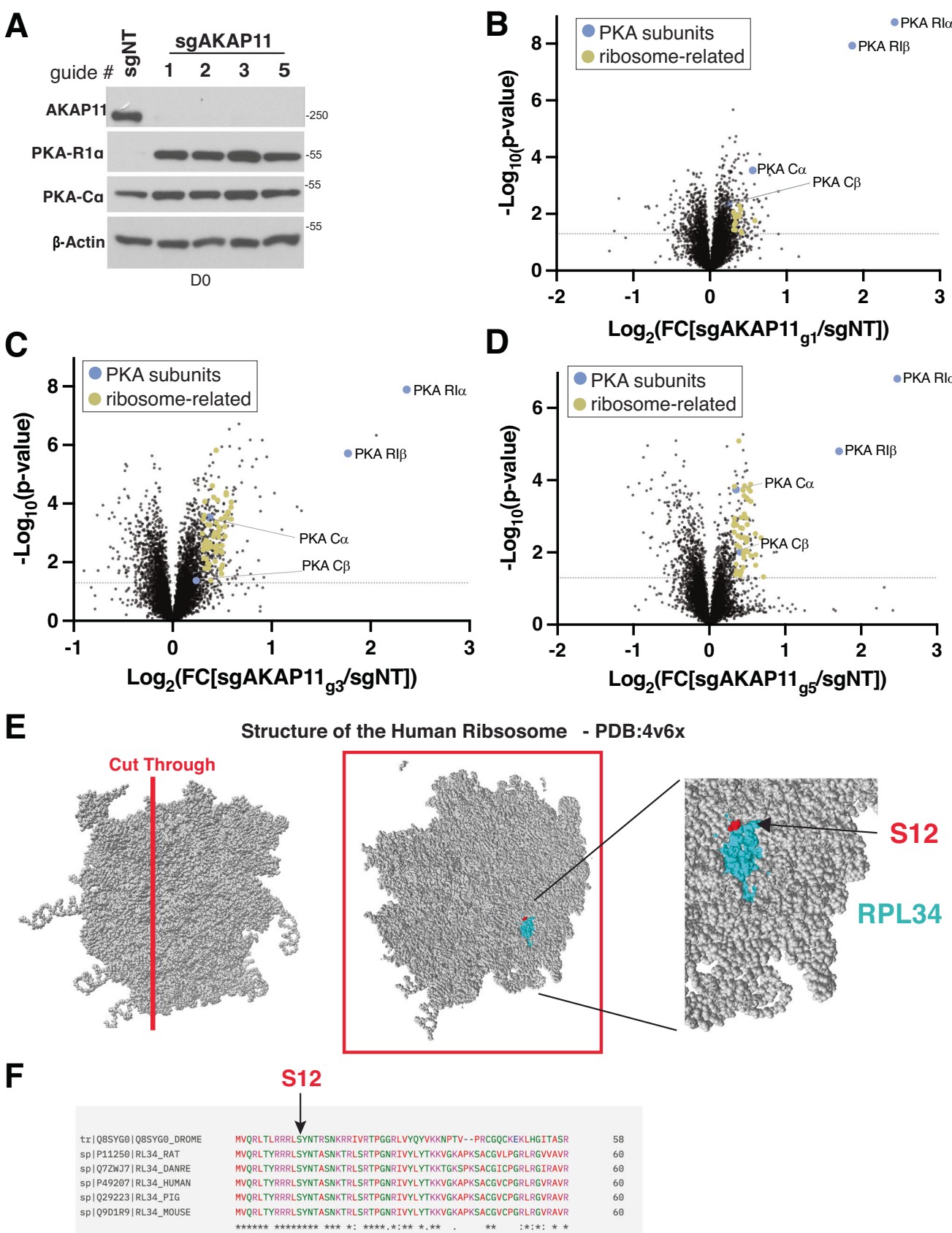

◀ **Figure EV4.   PKA holocomplex and ribosome related proteins are enriched in AKAP11-null i3 neurons.**

(A) Immunoblot of whole cell lysate of DIV0 i3 neurons validating degree of knockdown of four sgAKAP11 guides compared to sgNT. (B–D) Proteomic analysis of DIV7 i3 neurons in 3 different guides targeting AKAP11. $n = 3$ independent biological replicates for all conditions. *p*-values calculated using two-tailed unpaired *t* test. Blue nodes indicate PKA subunits. Gold nodes indicate ribosome-associated proteins (ribosome, cytoplasmic and ribosome biogenesis). (E) The cryoEM structure of the human ribosome with RPL34 (cyan) deeply buried in the complex and Ser12 annotated in red. (F) Sequence alignment of RPL34 across species showing conservations of Ser12.

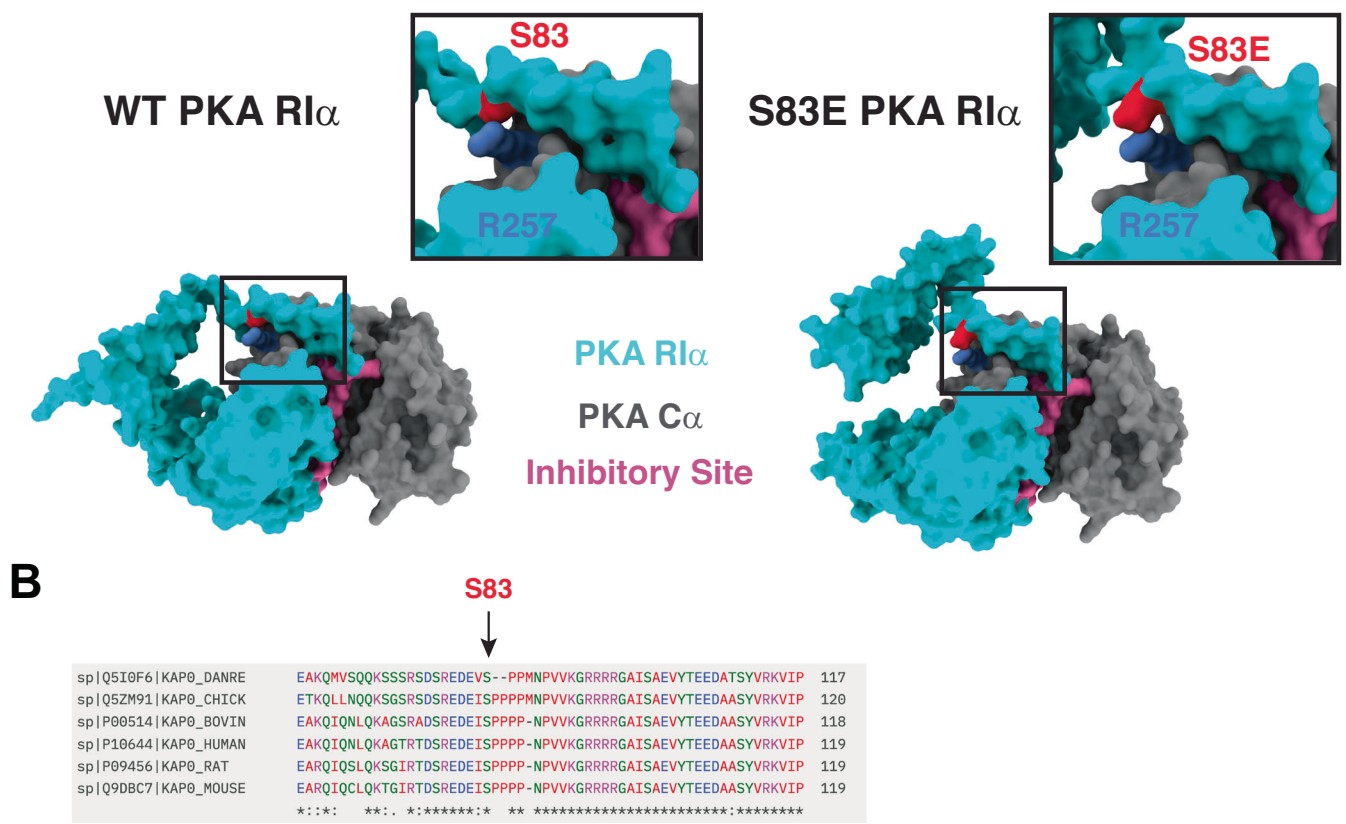

**A**

**AlphaFold PKA RIα - PKA Cα Complex Prediction**

WT PKA RIα

S83

R257

S83E PKA RIα

S83E

R257

PKA RIα
PKA Cα
Inhibitory Site

**B**

S83

```
sp|Q5I0F6|KAP0_DANRE   EAKQMVSQQKSSSRSDSREDEVS--PPMNPVVKGRRRRGAISAEVYTEEDATSYVRKVIP   117
sp|Q5ZM91|KAP0_CHICK   ETKQLLNQQKSGSRSDSREDEISPPPPMNPVVKGRRRRGAISAEVYTEEDAASYVRKVIP   120
sp|P00514|KAP0_BOVIN   EAKQIQNLQKAGSRADSREDEISPPPP-NPVVKGRRRRGAISAEVYTEEDAASYVRKVIP   118
sp|P10644|KAP0_HUMAN   EAKQIQNLQKAGTRTDSREDEISPPPP-NPVVKGRRRRGAISAEVYTEEDAASYVRKVIP   119
sp|P09456|KAP0_RAT     EARQIQSLQKSGIRTDSREDEISPPPP-NPVVKGRRRRGAISAEVYTEEDAASYVRKVIP   119
sp|Q9DBC7|KAP0_MOUSE   EARQIQCLQKTGIRTDSREDEISPPPP-NPVVKGRRRRGAISAEVYTEEDAASYVRKVIP   119

                       *::*:   **:. *:******:*  ** ********************:********
```

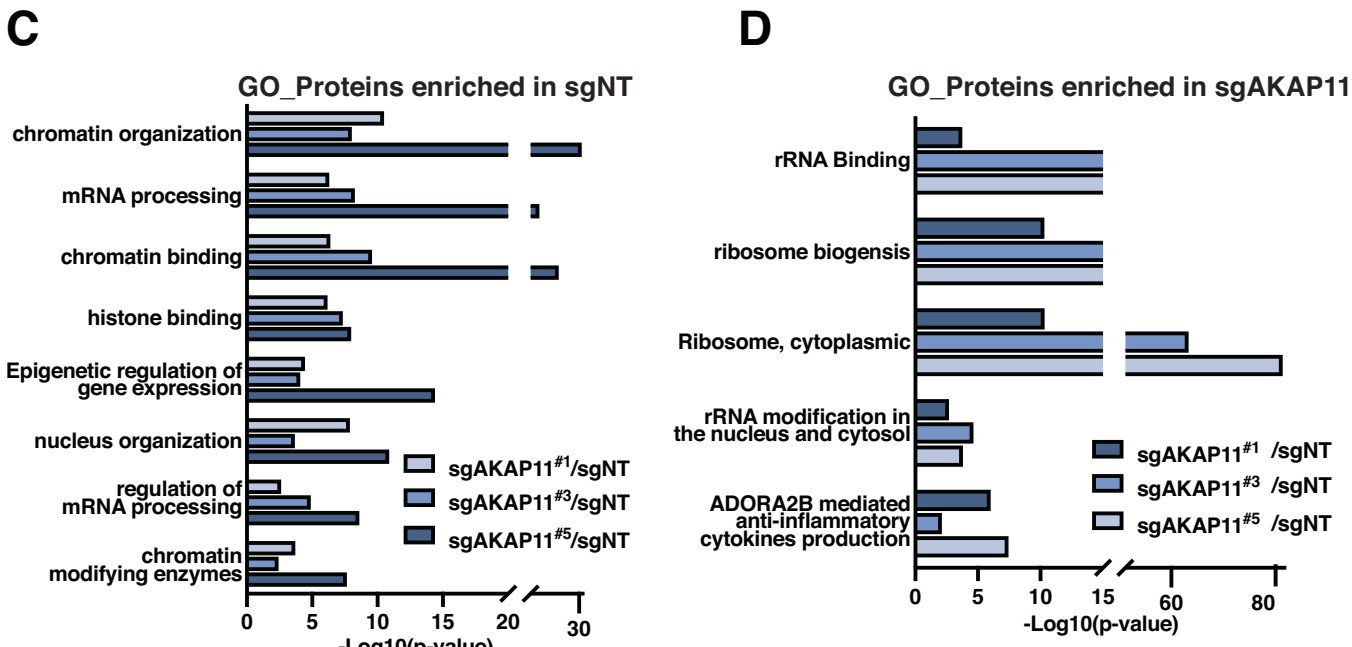

**C**

**GO_Proteins enriched in sgNT**

- chromatin organization
- mRNA processing
- chromatin binding
- histone binding
- Epigenetic regulation of gene expression
- nucleus organization
- regulation of mRNA processing
- chromatin modifying enzymes

sgAKAP11#1/sgNT
sgAKAP11#3/sgNT
sgAKAP11#5/sgNT

-Log10(p-value)

**D**

**GO_Proteins enriched in sgAKAP11**

- rRNA Binding
- ribosome biogensis
- Ribosome, cytoplasmic
- rRNA modification in the nucleus and cytosol
- ADORA2B mediated anti-inflammatory cytokines production

sgAKAP11#1 /sgNT
sgAKAP11#3 /sgNT
sgAKAP11#5 /sgNT

-Log10(p-value)

◀ **Figure EV5. Structural and conservation analysis of PKA RIα Ser83.**

(A) Alphafold prediction of the full length PKA holoenzyme complex, both WT and containing S83E mutant RIα. PKA RIα contains an inhibitory site (pink) that inserts in the active site of PKA Cα (gray). An unstructured loop nearby contains Ser83 (red). The S83E mutation, which mimics phosphorylation, is predicted to form a salt bridge with Arg257 of Cα and induce a significant conformational rearrangement in RIα. (B) Sequence alignment of PKA RIα across species showing conservation of Ser83. (C, D) Pathway enrichment analysis of DIV7 i3 neurons comparing (C) pathways enriched in sgNT (D) pathways enriched in sgAKAP11. Proteins included in the pathway enrichment analysis had a $-0.2 \geq \mathrm{Log_2FC} \geq 0.2$ and $P$ value < 0.05, two-tailed $t$ test.

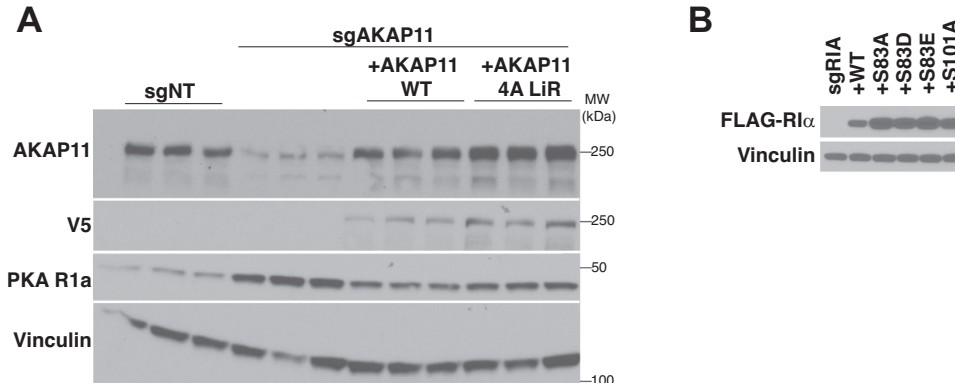

**Figure EV6. Validation of knockdown and rescue of AKAP11 and RIa.**

(A) Immunoblot of whole cell lysate from HEK293T sgNT and sgAKAP11 and sgAKAP11 cells stably expressing the indicated V5-AKAP11 construct in triplicate. Cells were grown in DMEM + 10% dFBS. These samples were used for phosphoproteomic experiments. (B) Western blots of sgRIα lysates used in Kemptide kinase assay showing vinculin as a loading control and transient expression of FLAG-RIα rescue constructs.

