## [Peer Review File · The EMBO Journal]

Autophagosomes anchor an AKAP11-dependent regulatory checkpoint that shapes neuronal PKA signaling

Ashley Segura-Roman, Y. Rose Citron, Myungsun Shin, Nicole Sindoni, Alex Maya-Romero, Simon Rapp, Claire Goul, Joseph Mancias, and Roberto Zoncu

Corresponding author(s): Roberto Zoncu (rzoncu@berkeley.edu)

Review Timeline:

Submission Date:	13th Aug 24
Editorial Decision:	28th Sep 24
Revision Received:	7th Feb 25
Editorial Decision:	2nd Mar 25
Revision Received:	13th Mar 25
Accepted:	24th Mar 25

Editor: William Teale

Transaction Report:

Dear Roberto,

Thank you again for the submission of your manuscript entitled "Autophagosomes enable an AKAP11-dependent regulatory checkpoint that shapes neuronal PKA signaling" and for your patience during the review process. We have now received the reports from the referees, which I copy below.

As you will see, referee #2 has significant concerns over the interpretation of data you present. Referees #1 and #3 request some further experimental work and textual clarification of key points. That said, both of these referees point out the timeliness and solidity of your work.

Based on the overall interest expressed in the reports, I would therefore like to invite you to address the comments of all referees in a revised version of the manuscript. In any revision, Referee #2 will have to be satisfied that AKAP11-PKA complex indeed initiates pS83/S83-dependent and location-specific downstream signalling events. I should add that it is The EMBO Journal policy to allow only a single major round of revision and that it is therefore important to resolve the main concerns at this stage. I believe the concerns of the referees are reasonable and addressable, but please contact me if you have any questions, need further input on the referee comments or if you anticipate any problems in addressing any of their points. Please, follow the instructions below when preparing your manuscript for resubmission.

I would also like to point out that as a matter of policy, competing manuscripts published during this period will not be taken into consideration in our assessment of the novelty presented by your study ("scooping" protection). We have extended this 'scooping protection policy' beyond the usual 3 month revision timeline to cover the period required for a full revision to address the essential experimental issues. Please contact me if you see a paper with related content published elsewhere to discuss the appropriate course of action.

Again, please contact me at any time during revision if you need any help or have further questions.

Thank you very much again for the opportunity to consider your work for publication. I look forward to your revision.

Best regards,

William

William Teale, Ph.D.
Editor
The EMBO Journal

When submitting your revised manuscript, please carefully review the instructions below and include the following items:

- 1) a .docx formatted version of the manuscript text (including legends for main figures, EV figures and tables). Please make sure that the changes are highlighted to be clearly visible.
- 2) individual production quality figure files as .eps, .tif, .jpg (one file per figure).
- 3) a .docx formatted letter INCLUDING the reviewers' reports and your detailed point-by-point response to their comments. As part of the EMBO Press transparent editorial process, the point-by-point response is part of the Review Process File (RPF), which will be published alongside your paper.
- 4) a complete author checklist, which you can download from our author guidelines ([https://wol-prod-cdn.literatumonline.com/pb-assets/embo-site/Author Checklist%20-%20EMBO%20J-1561436015657.xlsx](https://wol-prod-cdn.literatumonline.com/pb-assets/embo-site/Author%20Checklist%20-%20EMBO%20J-1561436015657.xlsx)). Please insert information in the checklist that is also reflected in the manuscript. The completed author checklist will also be part of the RPF.
- 5) Please note that all corresponding authors are required to supply an ORCID ID for their name upon submission of a revised manuscript.
- 6) We require a 'Data Availability' section after the Materials and Methods. Before submitting your revision, primary datasets produced in this study need to be deposited in an appropriate public database, and the accession numbers and database listed

under 'Data Availability'. Please remember to provide a reviewer password if the datasets are not yet public (see <https://www.embopress.org/page/journal/14602075/authorguide#datadeposition>). If no data deposition in external databases is needed for this paper, please then state in this section: This study includes no data deposited in external repositories. Note that the Data Availability Section is restricted to new primary data that are part of this study.

Note - All links should resolve to a page where the data can be accessed.

8) For data quantification: please specify the name of the statistical test used to generate error bars and P values, the number (n) of independent experiments (specify technical or biological replicates) underlying each data point and the test used to calculate p-values in each figure legend. The figure legends should contain a basic description of n, P and the test applied. Graphs must include a description of the bars and the error bars (s.d., s.e.m.).

9) We would also encourage you to include the source data for figure panels that show essential data. Numerical data can be provided as individual .xls or .csv files (including a tab describing the data). For 'blots' or microscopy, uncropped images should be submitted (using a zip archive or a single pdf per main figure if multiple images need to be supplied for one panel). Additional information on source data and instruction on how to label the files are available at .

10) We replaced Supplementary Information with Expanded View (EV) Figures and Tables that are collapsible/expandable online (see examples in <https://www.embopress.org/doi/10.15252/embj.201695874>). A maximum of 5 EV Figures can be typeset. EV Figures should be cited as 'Figure EV1, Figure EV2" etc. in the text and their respective legends should be included in the main text after the legends of regular figures.

12) Our journal encourages inclusion of *data citations in the reference list* to directly cite datasets that were re-used and obtained from public databases. Data citations in the article text are distinct from normal bibliographical citations and should directly link to the database records from which the data can be accessed. In the main text, data citations are formatted as follows: "Data ref: Smith et al, 2001" or "Data ref: NCBI Sequence Read Archive PRJNA342805, 2017". In the Reference list, data citations must be labeled with "[DATASET]". A data reference must provide the database name, accession number/identifiers and a resolvable link to the landing page from which the data can be accessed at the end of the reference. Further instructions are available at .

13) In order to increase the reproducibility and reach of your work, The EMBO Journal includes a table of reagents that were used in the study. Please provide this along with your revisions.

Further instructions for preparing your revised manuscript:

We realize that it is difficult to revise to a specific deadline. In the interest of protecting the conceptual advance provided by the work, we recommend a revision within 3 months (27th Dec 2024). Please discuss the revision progress ahead of this time with the editor if you require more time to complete the revisions. Use the link below to submit your revision:

Referee #1:

This is a well written and interesting article that utilizes proteomic-based analysis of immunopurified lysosomes to identify the A-kinase anchoring protein AKAP11-PKA holocomplex as a prominent autophagy-associated protein kinase complex. Cell based knockout of AKAP11 in induced pluripotent stem cell-derived neurons reveals dysregulation of multiple pathways for neuronal homeostasis. Other investigators have shown that AKAP11 is mutated in schizophrenia and bipolar disorders. This has led the authors to propose that AKAP11 anchored kinase complexes may be positioned for a possible mechanistic link to SZ/BP pathophysiology. Overall, this is a solid study with well controlled experiments. The findings are interesting and represent a valuable contribution. I have a few issues and suggestions that may need to be addressed.

1) The paper opens with compelling proteomic data. However, some of this data is difficult to interpret, superfluous to the story or hard to follow. Some relatively simple cosmetic changes could transform the interpretation of this important finding. Figure 1A outlining the fractionation procedure is too small. A more complete volcano plot for figure 1B could be valuable at this stage of the story (most data are in fig 2).

2) Schematic figure 1C should be placed in supplemental material. Figure 1D is overwhelming and rather small. It is impossible to read the names of the proteins in each cluster unless one really expands the font. Consider focusing on the AKAP11 sub complex and make it bigger.

3) What is the link between AKAP11 and AKAP5 in the sub complex presented in figure 1D Can both AKAPs be co-precipitated together?

4) (minor point) Can the authors comment on why they do not see any phase separation of R1 α in the IF data presented in Figure 1c. This contemporary phenomenon has been reported previously.

5) The experiments in figure 3 are interesting and well controlled. One possible addendum to this work would be to evaluate the impact of blocking AKAP11 associated PKA activity using the potent and highly selective compound BLU2864 an ATP-competitive PRKACA inhibitor (IC 50 =0.3 nM).

6) Could experiments in figure 4 include controls using a modified form of AKAP11 that is unable to anchor PKA holoenzymes. Would this mutant AKAP form have a similar or different effect as AKAP114ALIR?

7) The identification of Ser83 as a phosphosite on R1 α is very interesting. Is this site conserved across all R1 α and R1 β isoforms. Although biochemical experiments testing the effect of mutating the PKA anchoring on AKAP11 and measuring changes in R1 α ser 83 phosphorylation may be beyond the scope of this study. The authors should speculate in more detail about these effects in the discussion.

Referee #2:

The manuscript by Segura-Roman and colleagues examines the role of AKAP11 in regulating PKA during autophagy, with a particular focus on its function in neurons.

The involvement of AKAP11 in both autophagy and neuronal processes has been previously documented, with AKAP11 shown to control the homeostasis of the PKA-R1 α complex by acting as an autophagy receptor through its LIR domain (ref 24). Additionally, AKAP11's role in neurons has also been reported (ref 25), where it regulates levels of the holoenzyme (R1 α and C α) in an autophagy- and LC3-dependent manner, thereby influencing PKA-dependent signalling events.

In the present manuscript, the authors report that the co-localisation and transfer of R1 α to lysosomes require AKAP11. They also highlight that the entire R1 α and C α complex relies on AKAP11 and autophagy for its delivery to and degradation within lysosomes. This process is impaired in cells expressing the LIR-mutant form of AKAP11 and in ATG7-depleted cells. Regarding the role of AKAP11 in neurons, the authors investigate the phosphoproteomic profile in cells lacking AKAP11 and reconstituted with either wild-type or LIR-mutant AKAP11. They confirm that AKAP11 depletion leads to an increase in both R1 α and C α levels in neurons as well. The study identifies 228 differentially phosphorylated peptides, most of which are non-PKA-dependent phospho-sites, suggesting that AKAP11 depletion impacts PKA-independent processes. Among the 13 PKA-dependent phospho-sites, some exhibit increased phosphorylation while others show decreased levels in AKAP11-depleted cells, indicating that the absence of AKAP11 has a non-uniform effect on PKA substrate phosphorylation.

The authors identify the phospho-site S83 in R1 α as dependent on AKAP11, as its phosphorylation decreases in AKAP11-depleted cells but is restored in cells expressing V5-AKAP11. They demonstrate that S83 phosphorylation, by an unidentified kinase, regulates PKA activity. However, when they investigate whether AKAP11's regulation of S83 phosphorylation is linked to its role as an autophagy receptor, they find that this is not the case. In fact, phosphorylation is even more robustly restored by the LIR-mutant version of V5-AKAP11. In contrast, other altered phospho-sites, such as that of NCKAP5L, are only rescued by wild-type AKAP11 and not by the LIR-mutant form.

The simplest conclusion from these findings is that certain downstream events of AKAP11 depend on its function as an autophagy receptor, while others do not. The authors, however, bypass this hypothesis and propose a more intriguing, though speculative, model. In this model, the AKAP11-PKA complex in the cytosol exposes the S83 phospho-site to the unidentified kinase, whereas the same complex on the autophagosome conceals the S83 phospho-site.

Specific points

1. Fig. 2A: The authors observe a statistically significant increase in R1 α levels upon lysosomal protease inhibition, while the levels of C α are much less affected, both in terms of extent and statistical significance. The authors should address this discrepancy. One potential explanation is that R1 α might be more sensitive to lysosomal degradation than C α , possibly due to differential recruitment to autophagosomes. R1 α may also have additional functions that make it more susceptible to autophagy-mediated regulation. Alternatively, C α could be degraded through other pathways, such as the proteasome, rather than relying primarily on lysosomal proteases.

2. Fig. 2B: The authors compare the impact of ATG7 depletion only under conditions of lysosomal protease inhibition. However, they should also compare ATG7 knockout (KO) and control cells under steady-state conditions to more accurately assess the role of autophagy in regulating baseline PKA levels. Conducting this comparison under normal conditions would offer a clearer understanding of how autophagy contributes to the homeostasis of PKA without the influence of lysosomal stress or inhibition, providing insight into the basal role of autophagy in PKA regulation.

3. Fig. 3A: The total levels of AKAP11 in the input do not appear to be significantly affected by the inhibition of lysosomal degradation. This suggests that lysosomal degradation may not play a major role in the homeostatic regulation of AKAP11 levels, or that other compensatory mechanisms might be involved in maintaining its stability. The authors should address the extent to which lysosomal degradation contributes to AKAP11 regulation under various conditions. Additionally, they should consider whether alternative pathways, such as proteasomal degradation, could also be relevant for AKAP11 turnover, and discuss these potential mechanisms in their analysis.

Referee #3:

General Summary:

The manuscript by Segura-Roman and colleagues presents novel insights into Protein Kinase A (PKA) regulation through its localisation to the autophagosomal-lysosomal system. Moreover, the authors report that the scaffolding protein AKAP11, which is linked to schizophrenia and bipolar disorder, facilitates the interaction of PKA with autophagosomes via direct association with LC3B through a LIR motif. As a consequence, the autophagosomal interaction regulates PKA signalling by both facilitating its kinase activity and also via its eventual degradation, thereby providing spatial and temporal control of PKA activity and exposure to its substrates. While the manuscript focuses on mechanistically identifying how autophagy-specific localisation regulates PKA activity, it generates greater significance by investigating the mechanism in iNeurons, demonstrating a good level of physiological relevance that may be important for understanding the pathophysiology of psychological disorders.

Overall, this is a technically very well-designed and executed biochemical analysis, and the quality of the data is excellent. The manuscript is very well written, the narrative is very clear and draws very well on previously published works. Apart from one important technical query and some minor issues with the manuscript (see below), this study advances the field of PKA signalling by reporting a new level of regulation that may be significant for the molecular understanding of psychological conditions.

Major Concerns:

In Figure 1, panel E, (and Figure 4, panel E) AKAP11 is not detectable in NT without BafA1, but in Figure 3A, it is and seems to decrease in NT+BafA1, in opposition to Figures 1 and 4. This is an important control and ideally, all things being equal (sgNT and HEK293T cells), shouldn't both blots be reproducible? Could the authors please address this inconsistency, does it have any import on the results? The input lanes for both these figures also look very different.

Minor Concerns:

The fourth line of the abstract, degradation*

The last line of the abstract; modulates*

In the methods section, under LysoIP, the second sentence needs to be corrected for clarity. The following sentence requires a degrees symbol.

Cell lysis and co-IP section of methods require (w/v) and (v/v) for recipes, also missing in other sections.

In the phosphoproteomic MS section, the first sentence is missing a degrees symbol.

The final section of the methods mentions Fiji, not ImageJ which was used earlier, but requires further information on Fiji.

Referee #1:

This is a well written and interesting article that utilizes proteomic-based analysis of immunopurified lysosomes to identify the A-kinase anchoring protein AKAP11-PKA holocomplex as a prominent autophagy-associated protein kinase complex. Cell based knockout of AKAP11 in induced pluripotent stem cell-derived neurons reveals dysregulation of multiple pathways for neuronal homeostasis. Other investigators have shown that AKAP11 is mutated in schizophrenia and bipolar disorders. This has led the authors to propose that AKAP11 anchored kinase complexes may be positioned for a possible mechanistic link to SZ/BP pathophysiology. Overall, this is a solid study with well controlled experiments. The findings are interesting and represent a valuable contribution. I have a few issues and suggestions that may need to be addressed.

We thank the reviewer for acknowledging the significance and rigor of our findings.

1) The paper opens with compelling proteomic data. However, some of this data is difficult to interpret, superfluous to the story or hard to follow. Some relatively simple cosmetic changes could transform the interpretation of this important finding. Figure 1A outlining the fractionation procedure is too small. A more complete volcano plot for figure 1B could be valuable at this stage of the story (most data are in fig 2).

2) Schematic figure 1C should be placed in supplemental material. Figure 1D is overwhelming and rather small. It is impossible to read the names of the proteins in each cluster unless one really expands the font. Consider focusing on the AKAP11 sub complex and make it bigger.

We thank the reviewer for both these suggestions to present our data in a clearer way. We have made Figure 1A larger (**new Figure 1A**) and moved Figure 1C to the supplemental data figures (**new figure EV2**). Additionally, we made the AKAP11 sub complex in larger for clarity and emphasis (**new Figure 1C**).

3) What is the link between AKAP11 and AKAP5 in the sub complex presented in figure 1D Can both AKAPs be co-precipitated together?

In protein network diagrams, orange nodes indicate proteins that were hits in our lysolP, whereas blue nodes are proteins that bind to them based on the BioGrid protein-protein interaction database, and are meant only to provide context to the subcomplexes (**new Figure 1C, new figure EV2A**). Accordingly, AKAP5 was not detected in lysosome immunoprecipitates. Additionally, there is no line connecting it to AKAP11, indicating there is no physical interaction based on published datasets.

4) (minor point) Can the authors comment on why they do not see any phase separation of R1 α in the IF data presented in Figure 1c. This contemporary phenomenon has been reported previously.

We thank the review for addressing the interesting phenomenon of PKA R1 α phase separation. We did take these observations into consideration. However, unlike previously reported (PMID: 32846158), were unable to see convincing evidence of GFP-PKA R1 α condensates in cells . We

were only observed PKA RI α puncta upon treatment with BafA1, however these spots clearly colocalized with RFP-LAMP1, indicating autophagy-mediated delivery to lysosomes (**Reviewer Figure 1A**). We believe the phenomenon of PKA RI α phase separation may be specific to certain conditions and is not involved in autophagic capture of the PKA holocomplex.

Reviewer Figure 1 PKA RI α forms puncta under BafA1 and Torin Treatments. A. Live-Imaging of HEK293A that over-express GFP-PKA RI α and RFP-Lamp1 were treated with 3-hour 500nM BafA1 and 250nM Torin for 3 hours to simultaneously stimulate autophagy and block lysosomal degradation, before imaging for GFP and RFP via confocal microscopy.

5) The experiments in figure 3 are interesting and well controlled. One possible addendum to this work would be to evaluate the impact of blocking AKAP11 associated PKA activity using the potent and highly selective compound BLU2864 an ATP-competitive PRKACA inhibitor (IC 50 =0.3 nM).

We thank the reviewer for their suggestion that AKAP11-bound PKA could exert feedback regulation on its autophagic capture. However, this possibility seems unlikely to us because we already showed that stimulating PKA C α activation with Forskolin has no effect on the binding of either PKA RI α or AKAP11 to LC3B (**Figure 3D**).

6) Could experiments in figure 4 include controls using a modified form of AKAP11 that is unable to anchor PKA holoenzymes. Would this mutant AKAP form have a similar or different effect as AKAP114ALIR?

We thank the reviewer for this excellent suggestion. In doing this experiment, we deleted two PKA binding sites, AA 610-623 and AA 1633-1646, (Whiting et al 2015, Ref. 16 in main text) from AKAP11 (AKAP11 Δ S1S2) to eliminate the two predicted sites for PKA RI α binding. In this mutant, the LIR motif is intact. Consistently, AKAP11 Δ S1S2 co-localized with the lysosome to a higher degree than AKAP11 4ALIR mutant. However, lysosomal capture of AKAP11 Δ S1S2 was significantly lower than AKAP11 WT (**Reviewer Figure 2A, 2B, new Figure EV3B, EV3C**). This data is in good agreement with our previous lysolIP data comparing WT and PKA RI α -deleted

cells (**Figure 4E**), further supporting that AKAP11 binding to PKA R1 α is important for AKAP11 capture into the lysosome.

Reviewer Figure 2 PKA R1 α binding defective AKAP11 has decreased lysosomal localization **A.** Immunofluorescence from HEK293T sgAKAP11 cells stably expressing V5-AKAP11 WT, V5-AKAP11 4ALIR (WSNL>AAAA), or V5-AKAP11 Δ S1S2 (deleted AA 610-623 & 1633-1646) were treated with 500nM BafA1 or DMSO for 5 hours before fixing and immunostaining for V5 and LAMP2. 10 μ m scale bar. **B.** Quantification of V5 and LAMP2 colocalization from 5 non-overlapping fields; **p(adj.)= 0.0037, unpaired t test.

7) The identification of Ser83 as a phosphosite on R1 α is very interesting. Is this site conserved across all R1 α and R1 β isoforms. Although biochemical experiments testing the effect of mutating the PKA anchoring on AKAP11 and measuring changes in R1 α ser 83 phosphorylation may be beyond the scope of this study. The authors should speculate in more detail about these effects in the discussion.

Serine 83 in PKA R1 α is indeed conserved at least down to Zebrafish, and is also present in PKA R1 β (**Reviewer Figure 3A and Fig. EV5B**).

We thank the Reviewer for suggesting mutating the PKA R1 α anchoring site on AKAP11.

Phosphorylation of ser83 is dependent on AKAP11, thus mutating the PKA anchoring sites on

AKAP11 should also result in ser83 not being phosphorylated, similar to complete AKAP11 deletion (**Figure 6B**). We believe these experiments will fit best in follow-up work detailing the identification of the Ser83-phosphorylating kinase, which we are currently undertaking.

A

Reviewer Figure 3 Ser83 conserved in both PKA RI isoforms. A. Sequence alignment of Human PKA RI α and PKA RI β highlighting conservation of Ser83.

Referee #2:

The manuscript by Segura-Roman and colleagues examines the role of AKAP11 in regulating PKA during autophagy, with a particular focus on its function in neurons.

The involvement of AKAP11 in both autophagy and neuronal processes has been previously documented, with AKAP11 shown to control the homeostasis of the PKA-R1 α complex by acting as an autophagy receptor through its LIR domain (ref 24). Additionally, AKAP11's role in neurons has also been reported (ref 25), where it regulates levels of the holoenzyme (R1 α and Ca) in an autophagy- and LC3-dependent manner, thereby influencing PKA-dependent signalling events.

We thank the reviewer for pointing out how our work builds on recent reports on AKAP11, thus adding key knowledge on this increasingly important protein. We would like to emphasize that we are the first to report the association and degradation of the entire AKAP11-R1 α -Ca holoenzyme to autophagosomes, and to define the molecular rules that govern this capture. These mechanistic experiments, combined with the first phosphoproteomic analysis of AKAP11-deleted neurons, provide key conceptual advances and correct a potential misconception, as previous studies focused on PKA R1 α and proposed that bulk degradation of PKA R1 α , mediated by AKAP11 and sparing Ca, enhances overall PKA signaling.

In the present manuscript, the authors report that the co-localisation and transfer of R1 α to lysosomes require AKAP11. They also highlight that the entire R1 α and Ca complex relies on AKAP11 and autophagy for its delivery to and degradation within lysosomes. This process is impaired in cells expressing the LIR-mutant form of AKAP11 and in ATG7-depleted cells.

Regarding the role of AKAP11 in neurons, the authors investigate the phosphoproteomic profile in cells lacking AKAP11 and reconstituted with either wild-type or LIR-mutant AKAP11. They confirm that AKAP11 depletion leads to an increase in both R1 α and Ca levels in neurons as well. The study identifies 228 differentially phosphorylated peptides, most of which are non-PKA-dependent phospho-sites, suggesting that AKAP11 depletion impacts PKA-independent processes. Among the 13 PKA-dependent phospho-sites, some exhibit increased phosphorylation while others show decreased levels in AKAP11-depleted cells, indicating that the absence of AKAP11 has a non-uniform effect on PKA substrate phosphorylation.

The authors identify the phospho-site S83 in R1 α as dependent on AKAP11, as its phosphorylation decreases in AKAP11-depleted cells but is restored in cells expressing V5-AKAP11. They demonstrate that S83 phosphorylation, by an unidentified kinase, regulates PKA activity. However, when they investigate whether AKAP11's regulation of S83 phosphorylation is linked to its role as an autophagy receptor, they find that this is not the case. In fact, phosphorylation is even more robustly restored by the LIR-mutant version of V5-AKAP11. In contrast, other altered phospho-sites, such as that of NCKAP5L, are only rescued by wild-type AKAP11 and not by the LIR-mutant form.

The simplest conclusion from these findings is that certain downstream events of AKAP11 depend on its function as an autophagy receptor, while others do not. The authors, however, bypass this hypothesis and propose a more intriguing, though speculative, model. In this model, the AKAP11-PKA

complex in the cytosol exposes the S83 phospho-site to the unidentified kinase, whereas the same complex on the autophagosome conceals the S83 phospho-site.

Collectively, our data support a role for the autophagosome as a scaffold that regulates the association of regulatory factors and substrates to AKAP11 and its bound holoenzyme. As the Reviewer points out, this is especially evident in the case of NCKAP5L, where phosphorylation at S440 and S767 is inhibited by wild-type, but not LIR-mutated, AKAP11.

In the case of S83 of PKA R1 α , regulation by WT and LIR-mutated AKAP11 go in the same direction, which could also be consistent with an autophagy-independent role of AKAP11 in scaffolding the S83-phosphorylating kinase. However, our phosphoproteomic data show *even more* S83 phosphorylation in cells reconstituted with LIR-mutated AKAP11 over wild-type.

As independent evidence for the idea that autophagosomes may shield the holoenzyme from the S83-phosphorylating kinase, a recent manuscript (Zhou et al. 2024, Ref. 25 in main text), reported phosphoproteomic data from WT and ATG14 KD iNeurons. We replotted this data to create a volcano plot showing phospho-site enrichment in ATG14 KD relative to WT iNeurons. This data shows that S83 is among the most enriched phospho-sites in ATG14 KD iNeurons, again consistent with a possible role of autophagy in partially 'concealing' S83 of PKA R1 α from an upstream kinase (**Reviewer Figure 4A**).

Reviewer Figure 4 S83 phosphorylation is enriched in autophagy-deficient i3 Neurons A. Volcano plot showing phosphoproteomic analysis of whole-cell lysates from WT and ATG14 KD iNeurons. Data from Zhou et al., 2024.

However, we agree with the Reviewer that deciding between the two models would require extensive follow-up studies, involving the identification of the S83-phosphorylating kinase and reconstitution of this kinase reaction in vitro, which is clearly beyond the scope of the current manuscript. Thus, we have modified the Discussion and final model (**Figure 7**) to take into accounts both possibilities.

Specific points

1. Fig. 2A: The authors observe a statistically significant increase in R1 α levels upon lysosomal protease inhibition, while the levels of C α are much less affected, both in terms of extent and statistical significance. The authors should address this discrepancy. One potential explanation is that R1 α might be more sensitive to lysosomal degradation than C α , possibly due to differential recruitment to autophagosomes. R1 α may also have additional functions that make it more susceptible to autophagy-mediated regulation. Alternatively, C α could be degraded through other pathways, such as the proteasome, rather than relying primarily on lysosomal proteases.

We thank the reviewer for this critical observation. The simplest explanation for the higher degree of PKA R1 α degradation compared to PKA C α is that PKA R1 α is bound stably and constitutively to the autophagic receptor, AKAP11, whereas PKA C α is bound as an inverse function of the cAMP concentration (**Figure 3B, 3D**). Under standard growth conditions (complete media), PKA C α is clearly captured into lysosomes in an AKAP11-dependent manner, and is stabilized to a similar degree as PKA R1 α by BafA1 (**Figure 3A**), suggesting that PKA R1 α and PKA C α are equally susceptible to breakdown by lysosomal proteases.

2. Fig. 2B: The authors compare the impact of ATG7 depletion only under conditions of lysosomal protease inhibition. However, they should also compare ATG7 knockout (KO) and control cells under steady-state conditions to more accurately assess the role of autophagy in regulating baseline PKA levels. Conducting this comparison under normal conditions would offer a clearer understanding of how autophagy contributes to the homeostasis of PKA without the influence of lysosomal stress or inhibition, providing insight into the basal role of autophagy in PKA regulation.

We thank the reviewer for the suggestion, however without the use of lysosomal protease inhibitors we are unable to capture the contents of the lysosome, as they are being continually degraded. The use of protease inhibitors in conjunction with lysosome immunoprecipitation is crucial to discriminate between lysosomal resident proteins and autophagic cargo. As such, when we attempted to replot our ATG7 KO/WT data at baseline (DMSO) the results were uninterpretable. Again, we would like to emphasize that the standard in the field for assessing true lysosomal substrates is to measure the “flux” between baseline and proteolysis inhibited conditions.

3. Fig. 3A: The total levels of AKAP11 in the input do not appear to be significantly affected by the inhibition of lysosomal degradation. This suggests that lysosomal degradation may not play a major role in the homeostatic regulation of AKAP11 levels, or that other compensatory mechanisms might be involved in maintaining its stability. The authors should address the extent to which lysosomal degradation contributes to AKAP11 regulation under various conditions. Additionally, they should consider whether alternative pathways, such as proteasomal degradation, could also be relevant for AKAP11 turnover, and discuss these potential mechanisms in their analysis.

We thank the reviewer for this critical observation. The apparent decrease of AKAP11 in Figure 3A upon BafA1 treatment was a technical mistake that we did not catch. It has now been fixed and replaced (**Figure 3A**) showing the reproducible result that AKAP11 accumulates in the lysosome upon BafA1 treatment. Additionally, there has been precedent established by previous reports, which thoroughly examined the autophagy-dependent degradation of AKAP11, and our data agrees with what has been published. Furthermore, we found that AKAP11 degradation is more prominent in neuronal cells lines, as the expression is higher. In i3 Neurons, we show accumulation of AKAP11 over time upon BafA1 treatment (**Reviewer Figure 5A**). Additional support comes from replotting data from Hoyer et al. 2024 (PMID: 38429475), where day-12 WT or ATG12 KO iNeurons were harvested to survey protein accumulation in an autophagy-dependent manner. This data shows a pronounced accumulation of AKAP11, along with PKA R1 α and PKA C α , in autophagy-deficient neurons (**Reviewer Figure 5B**). Combined, these data suggest that autophagy is the main degradation mechanism for AKAP11.

Reviewer Figure 5 Degree of stabilization of AKAP11 and PKA holoenzyme in iNeurons. A. Immunoblot of i3 Neurons (DIV7) treated with 400nM BafA1 for the indicated time. **B.** Volcano plot of day-12 WT and ATG12 KO iNeuron total proteome using data from published study by Hoyer et al. 2024.

Referee #3:

General Summary:

The manuscript by Segura-Roman and colleagues presents novel insights into Protein Kinase A (PKA) regulation through its localisation to the autophagosomal-lysosomal system. Moreover, the authors report that the scaffolding protein AKAP11, which is linked to schizophrenia and bipolar disorder, facilitates the interaction of PKA with autophagosomes via direct association with LC3B through a LIR motif. As a consequence, the autophagosomal interaction regulates PKA signalling by both facilitating its kinase activity and also via its eventual degradation, thereby providing spatial and temporal control of PKA activity and exposure to its substrates. While the manuscript focuses on mechanistically identifying how autophagy-specific localisation regulates PKA activity, it generates greater significance by investigating the mechanism in iNeurons, demonstrating a good level of physiological relevance that may be important for understanding the pathophysiology of psychological disorders.

Overall, this is a technically very well-designed and executed biochemical analysis, and the quality of the data is excellent. The manuscript is very well written, the narrative is very clear and draws very well on previously published works. Apart from one important technical query and some minor issues with the manuscript (see below), this study advances the field of PKA signalling by reporting a new level of regulation that may be significant for the molecular understanding of psychological conditions.

We thank the reviewer for acknowledging the significance and rigor of our findings as well as pointing out the quality of our work.

Major Concerns:

In Figure 1, panel E, (and Figure 4, panel E) AKAP11 is not detectable in NT without BafA1, but in Figure 3A, it is and seems to decrease in NT+BafA1, in opposition to Figures 1 and 4. This is an important control and ideally, all things being equal (sgNT and HEK293T cells), shouldn't both blots be reproducible? Could the authors please address this inconsistency, does it have any import on the results? The input lanes for both these figures also look very different.

We thank the reviewer for this critical observation. The apparent decrease of AKAP11 in Figure 3A upon BafA1 treatment was a technical issue that we did not catch. We repeated the experiment and the new panel shows that, as predicted by our proteomics and imaging data, AKAP11 accumulates in lysates from BafA1-treated cells (**Figure 3A**).

Minor Concerns:

*The fourth line of the abstract, degradation**

*The last line of the abstract; modulates**

In the methods section, under LysolIP, the second sentence needs to be corrected for clarity. The following sentence requires a degrees symbol.

Cell lysis and co-IP section of methods require (w/v) and (v/v) for recipes, also missing in other sections.

In the phosphoproteomic MS section, the first sentence is missing a degrees symbol.

We thank the Reviewer for their careful reading and pointing out the above mistakes. They have now been fixed in the revised main text.

The final section of the methods mentions Fiji, not ImageJ which was used earlier, but requires further information on Fiji.

Fiji stands for “Fiji Is Just ImageJ”, i.e. it is the same software, but we have clarified this in the text.

Dear Roberto,

We have now received re-review reports from two referees, which I have included below. As you will see, you have addressed their concerns satisfactorily. Before I can finally accept the manuscript, there are some remaining editorial points which need to be addressed. In this regard would you please:

- include funding information in the Acknowledgments section and remove the separate "Funding" heading,
- include up to five keywords,
- use an alphabetical reference format; for longer author lists, list 10 authors + 'et al.',
- rename the conflict of interest statement as a "Disclosure and competing interests statement",
- refer to Figures EV5-EV6 as 'Figure' or 'Fig.' in the manuscript text,
- in the "Experimental study design and statistics" section of the author checklist, include the manuscript section referred to in the second positive response in the last (pink) column,
- update source file names, titles, legends and manuscript callouts to Dataset EV1-EV3,
- save the Appendix file in PDF format,
- include a 'Reagents and Tools' section,
- provide exact p values in the legends of figures 2D, F; 4D,
- state the statistical test used for data analysis in the legends of figures 2A, B, D, F; 5G, H; 6E, EV1 C, EV4 B, C, D; EV5 C, D,
- define 'n' in the legends of figures 2A, B, D, F; 4B, D; 6E; EV1 C, EV3 C, EV4 B, C, D,
- define error bars in the legends of figures 6E, EV3 C,
- define the length of the scale bar in figure 4C, and
- correct the section order as follows: Title page - Abstract & Keywords - Introduction - Results - Discussion - Methods - Data Availability - Acknowledgements - Disclosure and Competing Interests Statement - References - Figure Legends - Table(s) - Expanded View Figure Legends.

We include a synopsis of the paper (see <http://emboj.embojpress.org/>). Please provide me with a general summary image, a two sentence statement and 3-5 bullet points that capture the key findings of the paper.

I am looking forward to receiving your revised manuscript.

EMBO Press is an editorially independent publishing platform for the development of EMBO scientific publications.

Best wishes,

William

Yours sincerely,

William Teale, PhD
Editor
The EMBO Journal
w.teale@embojournal.org

See also figure legend guidelines: <https://www.embojpress.org/page/journal/14602075/authorguide#figureformat>

- a point-by-point response to the referees' comments, with a detailed description of the changes made (as a word file).
- a word file of the manuscript text.
- individual production quality figure files (one file per figure)
- a complete author checklist, which you can download from our author guidelines (<https://www.embojpress.org/page/journal/14602075/authorguide>).

- Expanded View files (replacing Supplementary Information)

We realize that it is difficult to revise to a specific deadline. In the interest of protecting the conceptual advance provided by the work, we recommend a revision within 3 months (31st May 2025). Please discuss the revision progress ahead of this time with the editor if you require more time to complete the revisions. Use the link below to submit your revision:

Referee #2:

The authors have adequately addressed the concerns raised in the previous review.

Referee #3:

I am still of the opinion that the manuscript is of high quality, and of interest to any readership in the proteostasis and neurobiology fields.

The authors have addressed my specific concerns satisfactorily, and upon reviewing the addressing of the two other reviewers comments I feel the authors have gone as far as possible in satisfying the reviewer's concerns.

I have no additional concerns arising, and no further suggestions for improving the manuscript.

All editorial and formatting issues were resolved by the authors.

Dear Roberto,

I am pleased to inform you that your manuscript has been accepted for publication in the EMBO Journal.

Congratulations to all involved!

Best wishes,

William

William Teale, PhD
Editor
The EMBO Journal
w.teale@embojournal.org
